# A 30 year monthly 5 km gridded surface elevation time series for the Greenland Ice Sheet from multiple satellite radar altimeters

Baojun Zhang[1], Zemin Wang[1†], Jiachun An[1], Tingting Liu[1†], Hong Geng[2]

[1]Chinese Antarctic Center of Surveying and Mapping, Wuhan University, Wuhan, 430079, China
[2]School of Resource and Environment Science, Wuhan University, Wuhan, 430079, China

[†]*Correspondence to*: Zemin Wang (zmwang@whu.edu.cn) and Tingting Liu (ttliu23@whu.edu.cn)

**Abstract.** A long-term time series of ice sheet surface elevation change (SEC) is an essential parameter to assess the impact of climate change. In this study, we used an updated plane-fitting least-squares regression strategy to generate a 30 year surface elevation time series for the Greenland Ice Sheet (GrIS) at monthly temporal resolution and $5 \times 5$ km grid spatial resolution using ERS-1, ERS-2, Envisat, and CryoSat-2 satellite radar altimeter observations obtained between August 1991 and December 2020. The ingenious corrections for intermission bias were applied using an updated plane-fitting least-squares regression strategy. Empirical orthogonal function (EOF) reconstruction was used to supplement the sparse monthly gridded data attributable to poor observations in the early years. Validation using both airborne laser altimeter observations and the European Space Agency GrIS Climate Change Initiative (CCI) product indicated that our merged surface elevation time series is reliable. The accuracy and dispersion of errors of SECs of our results were 19.3% and 8.9% higher, respectively, than those of CCI SECs, and even 30.9% and 19.0% higher, respectively, in periods from 2006–2010 to 2010–2014. Further analysis showed that our merged time series could provide detailed insight into GrIS SEC on multiple temporal (up to 30 years) and spatial scales, thereby providing opportunity to explore potential associations between ice sheet change and climatic forcing. The merged surface elevation time series data are available at http://dx.doi.org/10.11888/Glacio.tpdc.271658 (Zhang et al., 2021).

## 1 Introduction

Over recent decades, the Greenland Ice Sheet (GrIS) has experienced increasing substantial imbalance. Driven by atmospheric and oceanic warming (Straneo and Heimbach, 2013; Hanna et al., 2012), this imbalance has become a leading driver of global sea level change, whose contribution to which is about 0.42 mm yr$^{-1}$ (Shepherd et al., 2020) higher than the Antarctic Ice Sheet's about 0.30 mm yr$^{-1}$ (Shepherd et al., 2018). As a result of changes in surface mass balance (SMB) and ice dynamics, ice sheet elevation change (EC) is a direct indicator of climate change. Furthermore, with an appropriate density model for the snow and firn layer in addition to a model of the distribution of the ice layers within the firn column, EC can be used to monitor variation in ice sheet mass balance. Thus, a long-term time series of GrIS EC is essential to assess the impact of climate change on the GrIS (Sørensen et al., 2018). Since 1991, various satellite altimetry missions have made continuous observation of ice

sheet EC a reality (Shepherd et al., 2019; Simonsen et al., 2021). This approach, which uses measurements of surface EC (SEC) derived from satellite altimetry to monitor ice sheet mass balance, provides an unprecedented advantage in terms of spatiotemporal resolution in comparison with two other satellite-based techniques: gravimetric mass balance derived from satellite gravimetry and input–output balance derived from remotely sensed ice flow (Shepherd et al., 2019; Simonsen et al., 2021).

The effective life of a single satellite mission is limited, which means reconstruction of a long-term ice sheet elevation time series requires observations from multiple altimeter missions to be combined. In such a process, the method adopted to eliminate system biases is a crucial factor. System biases include intermission bias, ascending–descending bias, and time-variable penetration effects. It is generally believed that intermission bias is derived mainly from orbital errors, and differences in the centre of gravity and phase of antennae between satellites (Frappart et al., 2016). Owing to its distinct spatial pattern

(Zwally et al., 2005; Frappart et al., 2016), intermission bias is generally corrected for each grid cell using an estimate calculated from observations over overlapping epochs (Paolo et al., 2016; Sørensen et al., 2018; Adusumilli et al., 2018; Schröder et al., 2019; Shepherd et al., 2019; Simonsen et al., 2021). The ascending–descending bias can be considered to comprise both intra-mission ascending–descending bias and inter-mission ascending–descending bias (Zhang et al., 2020). Both are related to the angle between radar polarization and wind-induced features of the firn (Armitage et al., 2014; Remy et

al., 2006). The former can be corrected by introducing a term for satellite flight direction into a regression model (Simonsen and Sorensen, 2017; Mcmillan et al., 2014; Yang et al., 2019), or reduced by re-tracking the radar return waveform using a threshold re-tracker (Helm et al., 2014; Schröder et al., 2019). No specific treatment has been proposed for handling the latter, except that it is accounted for by the introduction of estimations of a series of parameters into a regression model (Zhang et al., 2020). It has been proven that using a large amount of surface elevation observations to fine-tune the correction of

intermission bias and ascending–descending bias can ensure better self-consistency and reliability of the combined time series of elevation (Zhang et al., 2020). Unfortunately, this method is unsuitable for combining data from multiple satellite missions simultaneously, because introduction into the fitting model of additional parameters and the increasingly complicated topological relationships between them will lead to regression failure. For mitigating time-variable penetration effects, there are currently three common approaches: including waveform parameters in the regression model (Flament and Remy, 2012;

Simonsen and Sorensen, 2017), re-tracking the radar return waveforms with a threshold re-tracker (Nilsson et al., 2016; Helm et al., 2014; Schröder et al., 2017), or applying a waveform deconvolution model to the radar return waveforms (Arthern et al., 2001; Mcmillan et al., 2016; Slater et al., 2019). However, none of these approaches can account completely for the time-variable penetration effects.

    The coverage of ground tracks of polar orbiting altimetry satellites over the polar ice sheets is uneven. Additionally, certain

outliers exist in altimeter observations, especially in relation to the early altimetry missions, e.g., ERS-1 (Schröder et al., 2019). These problems will result in lack of available data values in certain cells of a joint elevation time series. Thus, to estimate the volume or mass change over a basin or an entire ice sheet, gridding methods such as kriging (e.g., Bamber et al., 2009; Slater et al., 2018) , tension continuous curvature splines (e.g., Zhang et al., 2017) , or inverse distance weight (e.g., Chuter and

Bamber, 2015) are usually employed to interpolate or even extrapolate the results for to unobserved grid cells. However, such straightforward interpolations are unable to reflect the true patterns of elevation or EC in steep and very active areas across ice sheet margins (Hurkmans et al., 2012), not to mention the accuracy of the extrapolation results where there are insufficient constraints. Assuming that the spatial distribution pattern of the variation of ice sheet SEC is very small temporally, then orthogonal spatial maps of surface elevation (SE) variability can be extracted using empirical orthogonal function (EOF) decomposition from a sufficiently long elevation time series. Together with sparse observations, orthogonal spatial maps can be used to realize interpolation (reconstruction) of a time series of early satellite-derived SE. Actually, EOF reconstruction has already been used for reconstruction of sea surface temperature (e.g., Smith et al., 1996) and sea level change (e.g., Chambers et al., 2002; Jin et al., 2012) time series. The high-quality observations of Envisat and CryoSat-2, especially the higher-resolution coverage of Cryosat-2, provide potential for the use of EOF reconstruction for interpolation of an early elevation time series.

Here, we improve a previously proposed algorithm (Zhang et al., 2020) that requires a large volume of observations in an integrated adjustment model for simultaneous correction of intermission bias and ascending–descending bias. While retaining its advantages, we develop a 30 year (1991–2020) monthly $5 \times 5$ km gridded SE time series for the GrIS by merging ERS-1, ERS-2, Envisat, and CryoSat-2 radar altimetry observations. Then, to facilitate the use of the elevation time series, we use the EOF reconstruction method for more reliable interpolation of data for grid cells with missing values. In this paper, the details of the data processing are presented. The final merged SE time series dataset is freely available at http://dx.doi.org/10.11888/Glacio.tpdc.271658 (Zhang et al., 2021).

## 2 Material and methodology

### 2.1 Satellite radar altimetry data

In this study, we used ice sheet SE observations from four European Space Agency (ESA) satellite radar altimeter missions: ERS-1, ERS-2, Envisat, and CryoSat-2. Since the launch of ERS-1 in 1991, satellite radar altimeters have continuously collected SE observations for 30 years using similar Ku-band altimeters. Currently, following the retirement of the first three missions, only CryoSat-2 remains in operation.

For ERS-1 and ERS-2, we used the Level 2 (L2) Geophysical Data Record (GDR) product from the REAPER project version RP01, which has been reprocessed to align both the ERS and Envisat datasets (Brockley et al., 2017). For Envisat, we downloaded the latest L2 GDR product version 3.0 from ESA, which is better than the previous version (ver. 2.1) in terms of coverage and performance at cross-overs. For CryoSat-2, we used the latest Baseline D L2 GDR data provided by ESA. Over land ice, Baseline D improves the ascending–descending crossover statistics to 0.1 m from 1.9 m achieved with the previous version Baseline C (Meloni et al., 2020). Before performing combined calculations, all erroneous height records were eliminated using standard quality flags.

## 2.2 Airborne laser altimetry data

To bridge the gap in observations between the ICESat and ICESat-2 laser altimeter missions, the Operation IceBridge project (OIB) implemented more than 1000 airborne surveys during 2009–2020. During the OIB campaign, the airborne laser altimeter payload (i.e., the Airborne Topographic Mapper (ATM)) recorded a large number of high-precision ice sheet SE observations. Additionally, prior to OIB, several Pre-IceBridge airborne ATM surveys were conducted between 1993 and 2008. The OIB and Pre-IceBridge ATM SE datasets can both be download from the National Snow and Ice Data Center. Because their accuracy is 10 cm or better (Krabill et al., 2004), we used these ATM elevation measurements (i.e., ATM L2 product) to validate our merged SE time series. Additionally, SECs derived from OIB and Pre-IceBridge ATM measurements (i.e., ATM L4 product) were also used.

## 2.3 Generation of surface elevation time series

Our previous study (Zhang et al., 2020) demonstrated that using a large amount of data to fine-tune the correction of intermission bias and ascending–descending bias can develop a more self-consistent and reliable combined elevation time series. However, use of the updated plane-fitting least-squares regression model of Zhang et al. (2020) to merge data from three or more satellite missions is not straightforward. As the number of satellite altimetry missions involved in the calculation increases, additional terms of system bias and the increasingly complex topological relationships between them must be considered in the least-squares regression model, making the model overly complicated and ultimately incomprehensible. However, it is possible to divide the calculation into several individual steps, reducing the complexity of the model, while retaining its advantages.

First, the intra-mission ascending–descending bias at a grid cell for each radar altimeter mission can be estimated as follows:

$$
\begin{aligned}
h(lon_i, lat_i, t_i) = h_0 &+ a_0(lon_i - lon_0) + a_1(lat_i - lat_0) + a_2(lon_i - lon_0)^2 + a_3(lat_i - lat_0)^2 \\
&+ a_4(lon_i - lon_0)(lat_i - lat_0) + a_5(t_i - t_0) + a_6\cos\big(2\pi(t_i - t_0)\big) + a_7\sin\big(2\pi(t_i - t_0)\big) \\
&+ a_8\big(bs_i - \overline{bs}\big) + b_{AD}(-1)^{AD} + res(lon_i, lat_i, t_i),
\end{aligned} \tag{1}
$$

where $h(lon_i, lat_i, t_i)$ denotes the surface height measured at longitude $(lon_i)$, latitude $(lat_i)$, and time $(t_i)$, re-tracked by the ICE-1 re-tracker (Bamber, 1994) for ERS-1, ERS-2, and Envisat, the OCOG re-tracker (Wingham et al., 1986) for the CryoSat-2 LRM, and the Wingham/Wallis model fit re-tracker (Wingham et al., 2006) for the CryoSat-2 SARIn. As a threshold retracker, ICE-1 and OCOG are less sensitive to fluctuations in penetration, thereby being more precise in terms of ice sheet elevation measurements (Nilsson et al., 2016; Schröder et al., 2017; Slater et al., 2019). The reference epoch $t_0$ was set to 2010.0 in this study; $lon_0$, $lat_0$, and $h_0$ represent the longitude, latitude, and height (at $t_0$), respectively, of the centre of a grid cell; $a_0$–$a_4$ are the quadratic expansion for surface topography; $a_5$–$a_7$ denote the linear and seasonal signals for temporal changes of SE; $a_8$ is a parameter to mitigate the time-variable penetration effects of the radar signal using the anomaly of backscattered power $(bs_i - \overline{bs})$; $b_{AD}$ is for the intra-mission ascending–descending bias; $AD$ is assigned a value of 1 for ascending tracks or 0 for descending tracks; and $res(lon_i, lat_i, t_i)$ denotes the residuals of the regression. Note that the SARIn and LRM observations

of CryoSat-2 should be calculated separately here. The heights with the intra-mission ascending–descending bias corrected can be derived from

$$h^c(lon_i, lat_i, t_i) = h(lon_i, lat_i, t_i) - b_{AD}(-1)^{AD}. \tag{2}$$

Second, the intermission bias between Envisat and SARIn (or LRM) of CryoSat-2 can be calculated from the corrected heights $h^c(lon_i, lat_i, t_i)$:

$$\begin{aligned} h^c(lon_i, lat_i, t_i) = {} & h_0 + a_0(lon_i - lon_0) + a_1(lat_i - lat_0) + a_2(lon_i - lon_0)^2 + a_3(lat_i - lat_0)^2 \\ & + a_4(lon_i - lon_0)(lat_i - lat_0) + a_5(t_i - t_0) + a_6 \cos(2\pi(t_i - t_0)) + a_7 \sin(2\pi(t_i - t_0)) \\ & + a_8(bs_i - \overline{bs}) + b_{im}(-1)^{im} + res(lon_i, lat_i, t_i), \end{aligned} \tag{3}$$

where $b_{im}$ is for the intermission bias; $im$ is 1 for Envisat observations or 0 for CryoSat-2 observations. The correction of the intermission bias can be applied by

$$h^{cc}(lon_i, lat_i, t_i) = h^c(lon_i, lat_i, t_i) - b_{im}(-1)^{im}. \tag{4}$$

The above implies that Envisat is taken as reference, which means that the bias between SARIn and LRM will be corrected in this step. After the intermission bias corrected, Envisat and CryoSat-2 data will be consistent, and subsequently ERS-2 and

140 then ERS-1 data can also be corrected to be consistent with them.

Third, all the consistent heights $h^{cc}(lon_i, lat_i, t_i)$ can be used in the final least-squares regression to construct the SE time series:

$$\begin{aligned} h^{cc}(lon_i, lat_i, t_i) = {} & h_0 + a_0(lon_i - lon_0) + a_1(lat_i - lat_0) + a_2(lon_i - lon_0)^2 + a_3(lat_i - lat_0)^2 \\ & + a_4(lon_i - lon_0)(lat_i - lat_0) + a_5(t_i - t_0) + a_6 \cos(2\pi(t_i - t_0)) + a_7 \sin(2\pi(t_i - t_0)) \\ & + a_8(bs_i - \overline{bs}) + res(lon_i, lat_i, t_i). \end{aligned} \tag{5}$$

The elevation anomaly can be derived as follows:

$$\begin{aligned} \Delta h(lon_i, lat_i, t_i) = {} & h^{cc}(lon_i, lat_i, t_i) \\ & - \Big( a_0(lon_i - lon_0) + a_1(lat_i - lat_0) + a_2(lon_i - lon_0)^2 + a_3(lat_i - lat_0)^2 \\ & + a_4(lon_i - lon_0)(lat_i - lat_0) + a_5(t_i - t_0) + a_6 \cos(2\pi(t_i - t_0)) + a_7 \sin(2\pi(t_i - t_0)) \\ & + a_8(bs_i - \overline{bs}) \Big). \end{aligned} \tag{6}$$

Note that the removal of $h_0$ is to facilitate study of EC, and that $h_0$ can be used to generate an independent digital elevation model (DEM). The DEM and the corresponding surface slope and azimuth are shown in Fig. 1. When necessary, $h_0$ can be added back. Then, the monthly SE time series for a grid cell can be obtained as follows:

$$\overline{\Delta h}(lon_0, lat_0, t^j) = \frac{1}{n} \sum_{i=1}^{n} \Delta h(lon_0, lat_0, t_i), \tag{7}$$

where $n$ is the number of corrected elevations in month $t^j$.

To generate a robust time series of 5 km gridded elevations, the above least-squares fitting is first performed on a 2 km polar-stereographic grid over the GrIS using the ice sheet mask in Zwally et al. (2012). For each grid node, all observations within 2.5 km of the centre of the grid node are used for the iterative least-squares estimation under the constraints of 3σ outlier rejection criteria and the same thresholds as in Zhang et al. (2020). Then, a 40 km floating median low-pass filter, similar to Schröder et al. (2019), and the same spatiotemporal median filter as used by Zhang et al. (2020), are implemented to generate the final merged robust 5 km gridded time series.

## 2.4 Interpolation based on EOF reconstruction

Assuming that the spatial patterns of GrIS SECs are stationary in time, the three-dimensional GrIS SE anomaly time series $\nabla H(lon, lat, t)$ can be represented as a linear combination of the EOF modes $eof_i(lon, lat)$ and principal components $pc_i(t)$ (Chambers et al., 2002; Jin et al., 2012):

$$\nabla H(lon, lat, t) = \sum_{i=1}^{N} eof_i(lon, lat)pc_i(t) \, , \tag{8}$$

where $N$ is the total number of EOF modes; $lon$, $lat$, and $t$ denote the temporal and spatial position of a certain SE anomaly. The purpose of solving EOF modes is to supplement the sparse monthly gridded data attributable to poor observations in the early years. The average proportion of the monthly grid cells that need interpolations to be filled is 24.9% for ERS-1 and 7.4% for ERS-2, which are much higher than 1.1% for Envisat and 0.8% for CryoSat-2. In particular, there are 7 monthly grids with more than 63% of cells need to be interpolated during the ERS-1 period. Therefore, we use the gridded time series during 2003–2020 obtained from the higher quality observations of Envisat and CryoSat-2 here. To mitigate errors caused by extrapolation, only grid cells with at least 100 elevation anomalies in the 216 months of the 2003-2020 period are retained. The missing values in the gridded time series during 2003–2020 are interpolated using ordinary kriging, a technique usually employed to generate a DEM (Bamber et al., 2009; Slater et al., 2018).

Then, for the monthly grid that needs interpolation, the following equation can be established:

$$v_t = \sum_{i=1}^{M} eof_i(lon, lat)PC_i(t) - \text{T}(lon, lat, t) \, , \tag{9}$$

where $\text{T}(lon, lat, t)$ denotes the values already in this monthly grid; $M$ means choosing the first $M$ EOF modes; $v_t$ is the interpolation (reconstruction) error; and $PC_i(t)$ is the principal components to be estimated corresponding to each of the $M$ modes for this monthly grid, which can be estimated to minimize $v_t$ using a linear least-squares estimator. To determine $M$, we experimented by adjusting it from 1 to 216 modes. We found that both the percentage of the explained variance and the root mean square (RMS) difference between the reconstructed time series of monthly EC and that of the observations become insensitive after 30 modes with only minor changes, as can be seen in Fig. 2. Thereby, the elevation anomalies missing from grid cells during the period of the ERS missions are interpolated using EOF reconstruction. Note that we first deduct the

seasonal signals using a least-squares fitting model with a second-order polynomial and seasonal terms before performing the EOF reconstruction, and then add them back to the EOF reconstruction results here. The performance of both EOF reconstruction and ordinary kriging is shown in Fig. 3, illustrating the superiority of the former in comparison with the latter. Especially in extrapolation, there are many obvious over-interpolations in the ordinary kriging result.

Volume change of an ice sheet is an important parameter for determining the response of the ice sheet to the effects of climate change. The altimetric volume time series can be derived from the gridded SE time series as described in Zhang et al. (2020). Firstly, the effects of Glacial Isostatic Adjustment (GIA) and elastic solid earth rebound should be corrected, for they don't reflect changes due to ice and snow. Then, the altimetric volume anomaly for each cell can be obtained by multiplying the corrected SE anomaly by the area of the cell. The altimetric volume anomalies for individual drainage basins and major sectors can be calculated by integrating the resulting altimetric volume anomalies over larger regions. In this study, the ICE-6G_D model (Peltier et al., 2018) and a scale factor (Groh et al., 2012) were used to correct for the vertical crustal deformation related to GIA and elastic solid earth rebound.

## 2.5 Uncertainty for surface elevation time series

As described in Sect. 2.3, the elevation anomaly in a grid cell of the merged SE time series is derived using a median estimator. Thus, to obtain a realistic estimation of error, we also use the scaled median absolute deviation $MAD_S$ as a metric of its uncertainty following Ewert et al. (2012):

$$MAD_S = k \; median(|H - median(H)|). \tag{8}$$

The scale factor $k$ is set to 1.4826 to make $MAD_S$ a consistent estimator similar to the standard deviation.

As for those interpolated elevation anomalies in the gridded time series, because of the complicated interpolation methods adopted, it is difficult to estimate their uncertainty using formal error propagation techniques. Here, we use the scaled RMS of the residuals $\varepsilon$ derived from the elevation anomalies $h$ in a grid cell as follows:

$$h = b_0 + b_1 t + b_2 t^2 + b_3 \cos\big(2\pi(t - t_0)\big) + b_4 \sin\big(2\pi(t - t_0)\big) + \varepsilon, \tag{9}$$

where $b_0$ is a constant; and $b_1$–$b_4$ denote the linear, quadratic, and seasonal signals of the temporal changes of SE, respectively. A scale factor of 1.05 is used to compensate for the reduced RMS error due to fitting (Wahr et al., 2006).

It should be noted that when using our elevation time series to estimate the volume change for individual drainage basins and major sectors, the correlated uncertainties between adjacent grid cells should also be considered. According to Schröder et al. (2019), applying a scaling factor to the squared uncertainties can account for the autocorrelation over an area.

## 3 Results

### 3.1 Surface elevation anomaly time series

The average rate of SEC in a certain time interval can be calculated from SE time series using a least-squares fitting model
with a first-order polynomial and a sine wave with a 1 year period. The additional annual items are used to avoid the bias
caused when the entire annual cycle is not fully covered. Fig. 4 shows the climatological seasonal maps and the amplitude of
annual cycle of SE anomaly over the GrIS. The spatial distribution patterns and magnitude of the seasonal changes in SE of
the GrIS are clearly presented. The significant signals of seasonal variation are mainly concentrated in the ablation zone below
the equilibrium line identified in Mcmillan et al. (2016). Thinning in autumn (July-August-September) and thickening in spring
(January-February-March) are driven by the seasonal fluctuations in surface melting, snowfall and ice dynamics (Bartholomew
et al., 2011; Slater et al., 2021). Between May and August, surface melting and enhanced ice dynamics when the surface
meltwater gains access to the ice–bed interface, lubricating basal motion lower the surface in the ablation zone. Snowfall and
slowing in ice dynamics in September–April thicken the ice sheet. No evident seasonal fluctuations are found in the elevation
of the GrIS interior.

The average SEC rates and their uncertainties over the periods 1991–2000, 2001–2010, 2011–2020, and 1991–2020 from our
merged elevation anomaly time series are shown in Fig. 5. As reported by Shepherd et al. (2020), the GrIS has been losing ice
throughout most of the intervening period. Thus, maps of these long-term average SEC rates show signals of continuous
thinning in many areas along the coast. Overall, the most notable signals of GrIS thinning are concentrated on the west coast
of Greenland, especially along Melville Bay and near Jakobshavn Isbræ. Comparison of the average rates in the different
periods reveals significantly accelerated and expanded thinning in many outlet glaciers, e.g., Jakobshavn Isbræ and Upernavik
Isstrøm on the west coast of the GrIS, Zachariæ Isstrøm and Nioghalvfjerdsfjorden glacier in the northeast of the GrIS,
Kangerdlugssuaq Gletscher and Helheimgletscher in the southeast of the GrIS, and Petermann Gletscher and Humboldt
Gletscher in the northwest of the GrIS. The main contributor to the significant thinning detected in these regions is ice dynamics.
The volume of solid ice being discharged into the ocean is increasing because of warmer air and ocean temperatures (Mouginot
et al., 2015; Aschwanden et al., 2016; Shepherd et al., 2020; Wood et al., 2021). Signs of thickening are evident mainly in
accumulation areas with higher elevation in central and northwestern parts of the GrIS, e.g., the area near King Christian X
Land. These weak signals of thickening mainly reflect the increase in SMB caused by a combination of high snowfall and low
surface melting (Simonsen et al., 2021). These thinning and thickening spatial patterns are also confirmed from ICESat and
ICESat-2 (Smith et al., 2020; Ewert et al., 2012).

Irrespective of whether thickening or thinning, it can be seen that the rates of SEC in different periods vary. To gain insight
into the spatiotemporal changes of average SEC rates, the average SEC rates and their uncertainties at 5 year intervals during
1991–2020 for our time series are illustrated in Fig. 6. Considerable variation in the mean SEC rates is evident, e.g., abnormal
accumulation during 1996–2000, gradually increasing loss from 2000–2005 to 2011–2015, and deceleration of thinning during
2015–2020. Benefitting from the higher temporal and spatial resolution of our combined time series, the small-scale

spatiotemporal evolution of the average rates of SEC can be analysed in detail. Taking Jakobshavn Isbræ as an example, Fig. 6 clearly reveals its evolution from thinning in the early 1990s, to equilibrium in the late 1990s, to accelerated thinning in the first decade of the 2000s, and then to deceleration of thinning in recent years. Similarly, the evolution of other glaciers can be obtained from our time series. It should be noted that due to the natural defect of radar altimeter, our time series is not suitable for glaciers that are too small or too steep. Sørensen et al. (2018) has arbitrarily excluded all grid cells which are located on

slopes exceeding 1.5° to avoid the possible large uncertainty.

Our 5 km gridded time series can also provide a more detailed evolution of SEC characteristics on temporal scales of up to 30 years. The four examples presented in Fig. 7 illustrate that our results have the capability to pinpoint such signals of GrIS SEC. Jakobshavn Isbræ is the largest and fastest outlet glacier on the west coast of Greenland; however, its thinning throughout the observational period since 1991 is not continuous. For example, short-term deceleration of thinning and thickening during

1996–2001 and since 2014, caused by deceleration of the ice flow (Joughin et al., 2004; Khazendar et al., 2019), can be seen in Fig. 7(a). A rapid drop in the surface elevation of Jakobshavn Isbræ is evident during 2003–2013. The rate of surface lowering increases with increasing distance from the grounding line. During this period, the mean rate at position A (Fig. 7(a)) is up to $-2.85 \pm 0.04$ m yr$^{-1}$. Upernavik Isstrøm consists of five glaciers, all of which flow into the same fjord. Zachariæ Isstrøm and Nioghalvfjerdsfjorden drain the majority of the Northeast Greenland Ice Stream. Unlike Jakobshavn Isbræ, their

surfaces have lowered consistently since 1991 (Fig. 7(b) and (c)). The average rates of thinning at A, B, and C in Upernavik Isstrøm are $-1.34 \pm 0.03$, $-1.01 \pm 0.01$, and $-0.60 \pm 0.01$ m yr$^{-1}$, respectively. The thinning rates of Zachariæ Isstrøm and Nioghalvfjerdsfjorden are slower, i.e., mean SEC rates of $-0.67 \pm 0.02$, $-0.28 \pm 0.01$, and $-0.21 \pm 0.01$ m yr$^{-1}$ at A, B, and C, respectively. Similarly, the closer to the grounding line, the faster the rate of thinning of the ice of those glaciers. King Christian X Land, located in the northeast of the GrIS, is a highly representative accumulation area. Ice velocity in this area is very small

and there is no outflow glacier, and its change is driven mainly by SMB (Aschwanden et al., 2016; Velicogna et al., 2014). It shows weak continuous thickening over the entire observational period since 1991 (Fig. 7(d)). The rate of thickening at A, B, and C is $0.13 \pm 0.01$, $0.11 \pm 0.01$, and $0.06 \pm 0.01$ m yr$^{-1}$, respectively. Whether continuous thinning or thickening, the rates do not remain constant, i.e., there are always periods of acceleration or deceleration that are evident traces of the driving force of climate change on ice sheet change. For example, abnormal melting in 2012 and accumulation since 2016–2017, both driven

by the North Atlantic Oscillation (NAO), are clearly visible in the time series of the above regions (Wood et al., 2021; Simonsen et al., 2021).

**3.2 Ice sheet volume time series**

The volume time series of the entire GrIS and certain sub-regions estimated from our time series are shown in Figs. 7 and 8, respectively. Linear and quadratic trend estimates can be inferred from the volume time series using a least-squares fitting

model with a second-order polynomial and a sine wave with a 1 year period.

Over the entire GrIS, we detect an overall volume loss of $53.8 \pm 4.5$ km$^3$ yr$^{-1}$ with an acceleration in loss of $2.2 \pm 0.3$ km$^3$ yr$^{-2}$ during 1991–2020 (Fig. 8). Six of eight ice drainage systems show volume loss (Fig. 9). The largest volume loss ($19 \pm 1.4$ km$^3$

yr$^{-1}$) and greatest acceleration in loss ($0.9 \pm 0.1$ km$^3$ yr$^{-2}$) are both from the ice sheet along the northwestern coast (Fig. 9h). Drainage systems located in central western and southwestern parts of the GrIS are the other two largest contributors to ice loss with volumes and rates of acceleration of $-10.2 \pm 1.3$ km$^3$ yr$^{-1}$ and $-0.5 \pm 0.1$ km$^3$ yr$^{-2}$ (Fig. 9g), and $-10.2 \pm 1.9$ km$^3$ yr$^{-1}$ and $-0.6 \pm 0.1$ km$^3$ yr$^{-2}$ (Fig. 9f), respectively. The only two drainage systems to show volume accumulation are located in central eastern (Fig. 9c) and northeastern (Fig. 9b) parts of the GrIS. However, their trends of volume accumulation are very weak, i.e., $1.3 \pm 0.8$ km$^3$ yr$^{-1}$ and $0.1 \pm 0.1$ km$^3$ yr$^{-2}$, and $-1.6 \pm 2.6$ km$^3$ yr$^{-1}$ and $0.5 \pm 0.2$ km$^3$ yr$^{-2}$, respectively.

In addition to studying the long-term trend of altimetric volume change of the ice sheet, our merged time series also provides detailed insight into small-scale fluctuations in volume change that reflect the effects of climate change on a temporal scale of up to 30 years. The evolution of ice sheet volume for the entire GrIS and certain sub-regions can be divided into different processes, as shown in Figs. 7 and 8, respectively. Before 1997, because of the contribution of various drainage systems in western Greenland (Fig. 9(a) and (e)–(h)), the GrIS presented rapid volume loss. Simonsen et al. (2021) thought that these ice losses were attributable mainly to the main outflow glaciers along the west coast. Then, the overall volume of the GrIS was alleviated, as also confirmed by the changes in the 5-year average SEC rates (see Fig. 7). This is attributed to increase of the SMB in the northeastern and central eastern drainage systems (Fig. 9(b) and (c)), and to reduction of ice discharge of outlet glaciers along the west coast (Fig. 9(g) and (h)). Subsequently, the GrIS entered a period of rapid ice loss because of the reduced SMB that was mostly attributable to meltwater runoff and increased ice discharge (Fig. 9(a) and (e)–(h)) (Simonsen et al., 2021; Shepherd et al., 2020; Velicogna et al., 2014). Then, all drainage systems entered another period of slowdown in ice loss. In fact, these processes are full of the traces of the effects of climate change. The rapid ice loss since 2003 was driven by the transition of the NAO from a high positive phase to a low-to-negative phase, which reduced SMB by enhancing melting and reducing snowfall, and accelerated ice discharge of outlet glaciers by driving warmer subsurface waters on the continental shelf (Bevis et al., 2019; Wood et al., 2021). The subsequent slowdown was because the NAO transitioned back to a more positive phase. It came from the anomalous increase in snowfall and anomalously low surface melting due to NAO-driven shifts in atmospheric forcing since 2016–2017 (Shepherd et al., 2020; Simonsen et al., 2021) and the slowed ice discharge attributable to NAO-driven shifts in oceanic forcing since 2010 (Wood et al., 2021). The weak signal of volume accumulation of drainage systems located in central eastern and northeastern parts of the GrIS (Fig. 9(b) and (c)) was also attributed to the two short-term abnormally increased snowfalls driven by the shift of the NAO, one in the early 2000s (Shepherd et al., 2020) and the other in the late 2010s (Simonsen et al., 2021). The volume of accumulated low-density snow exceeded that of lost high-density ice.

## 4 Comparison to independent datasets

### 4.1 Comparison with airborne laser altimetry elevation

To validate our merged results, we first used the high-precision ATM L2 surface heights. Before performing a comparison, a 40 km floating median low-pass filter was applied to the ATM L2 data to eliminate outliers. Moreover, the mean height (at $t_0$)

of the centre of each grid cell $h_0$ was first added back into our merged GrIS SE anomaly time series to match the surface heights. Then, we searched for all ATM L2 observations located within 2.5 km and a 10 day interval of each of the grid nodes of our three-dimensional time series. The result of subtracting the elevation value of a grid node from the median of the ATM L2 observations represents the difference for that location. The results of the validation are shown in Fig. 10(a). It can be seen that the larger differences are concentrated primarily in steeper areas at the margins of the GrIS. This might be due to the poor

observation accuracy of radar altimeters in areas of complex terrain (Zhang et al., 2020). Another possible reason is that many of them are interpolations or extrapolations. Over the GrIS, the median, RMS error, and the 10th and 90th percentile ranges ($P_{90}-P_{10}$) are −2.82, 99.43, and 145.25 m, respectively (Table 1), i.e., better than obtained for the 5 km interpolated grid cells of the DEM of Slater et al. (2018), which are comparable to our calculations in terms of strategy and resolution (their median and RMS error values were 25.4 and 138.6 m, respectively).

**4.2 Comparison with airborne laser altimetry elevation changes**

We also used ATM L4 SECs to evaluate our merged results. Similarly, a 40 km floating median low-pass filter was also used to eliminate outliers in the ATM L4 SECs before performing the validation. The ATM L4 SECs are derived from every two coincident ATM elevation measurements. We compared the ATM L4 data points with grids in our merged gridded time series that lay within a 2.5 km radius and a 15 day interval of the observation instants of that point. Subsequently, the SE differences

and SEC differences between the ATM L4 observations and our merged time series at the same epochs were obtained, as shown in Fig. 10(b) and (c). The spatial distribution patterns of SE differences and SEC differences are similar to those of the SE differences mentioned in Sect. 4.1. The larger differences are distributed in areas with complex terrain at the margins of the GrIS. The median, RMS error, and $P_{90}-P_{10}$ over the GrIS are also listed in Table 1. Overall, the median values of these difference are both near 0, and the two $P_{90}-P_{10}$ values are both relatively small. Thus, although the RMS error of the SE

differences is larger than that of both Schröder et al. (2019) and Zhang et al. (2020) for the Antarctic Ice Sheet, our result is still considered reliable. Furthermore, the integrity of ATM L4 data covering only the outlet glaciers of the West Antarctic Ice Sheet and the Antarctic Peninsula Ice Sheet is limited.

**4.3 Comparison with ESA GrIS Climate Change Initiative elevation changes**

The ESA GrIS Climate Change Initiative (CCI) project has provided a dataset of SECs over the GrIS with a 5 year mean during

1992–2020 derived from ESA's Ku-band radar satellite level-2 data products, which can be downloaded for free from http://products.esa-icesheets-cci.org/products/details/cci_sec_2020.tar.gz/. Here, our results are verified through intercomparison with that dataset. For consistent comparison, 5 year average SEC rates for the same observation epochs were estimated from our time series using a least-squares fitting model with a first-order polynomial and a sine wave with a 1 year period. Then, we compared the CCI SECs or our SECs with ATM L4 SECs at each grid node located within a 2.5 km radius.

To remove the influence of interpolation using EOF reconstruction, we only compared results that were not interpolated. The median was also used to eliminate the influence of outliers.

Figure 10 shows the median, RMS error, and P$_{90}$−P$_{10}$ of the results of the intercomparison over the GrIS at 5 year intervals. It can be seen that our results are better in most periods, especially in those time intervals across the period of overlapping observations of Envisat and CryoSat-2 (i.e., from 2006–2010 to 2010–2014). Statistics of validation with GrIS CCI SECs listed in Table 2 also confirm this assertion. In comparison with the CCI SECs, the accuracy (RMS error) and dispersion of errors (P$_{90}$−P$_{10}$) of our results are improved by 19.3% and 8.9%, respectively, over all periods. In all periods from 2006–2010 to 2010–2014, the accuracy (RMS error) and dispersion of errors (P$_{90}$−P$_{10}$) of our results are improved by 30.9% and 19.0%, respectively. It might indicate that the effectiveness of our method for inter-mission bias correction for ERS-1 and ERS-2 has been reprocessed to align with Envisat by REAPER (Brockley et al., 2017).

## 4.4 Limitations of the merged surface elevation time series

Although a series of processes to ensure the accuracy and reliability of the merged results have been proved to be effective by comparison with other independent datasets above, there still exist some limitations in the merged SE time series. These limitations mainly come from the natural defects of radar altimeter.

The first is the penetration of the signal into the surface snow, which causes a radar altimeter not to observe the actual surface height of the ice sheet. Furthermore, surface processes such as melting, refreezing, and firn compaction might produce a new reflecting surface that could result in errors. For example, the abrupt increase in the CryoSat-2 recorded elevation in the interior the GrIS during the extreme melt event in July 2012 was resulting from the change of penetration depth caused by surface melting (Nilsson et al., 2015; Mcmillan et al., 2016). This study used elevations retracked by threshold offset center of gravity retracker (ICE-1 re-tracker and OCOG re-tracker) and the common strategy of including corrections for waveform parameters into the least-squares regression model (see Eq. (1)) to mitigate the time-variable penetration effects of the radar signal. Because it is less sensitive to changes in volume scattering, the threshold offset center of gravity retracker has been used to reduce the effect of penetration (Nilsson et al., 2015; Schröder et al., 2017; Schröder et al., 2019). The latter has also been performed in many previous studies (Flament and Remy, 2012; Sørensen et al., 2018; Zhang et al., 2020). However, as presented by Slater et al. (2019), the influence of the time-variable penetration depth would not be completely eliminated, even when applying a waveform deconvolution procedure (Mcmillan et al., 2016). Thereby, a small residual signal caused by the 2012 melt event and manifesting as a surface elevation increase signal is found in the merged time-series. In regions above 2000 m in altitude, the elevation increased by approximately 0.16 m on average between the months before (January–June, 2012) and after (August–December, 2012) the extreme melt event, consistent with pervious findings (Slater et al., 2019). In the future, with the accumulation of long-term continuous observations by satellite laser altimetry ICESat-2, it seems feasible to obtain actual penetration depth and model predictions to better compensate for the fluctuations in penetration depth. On the bright side, surface penetration suppresses noise induced by seasonal snowfall, making radar altimetric measurements more relevant to mass change than those obtained from laser altimetry (Sørensen et al., 2018). Therefore, our multiple radar altimetry missions SE time series is more suited to track dynamical processes and inter-annual or long-term surface processes (Zhang et al., 2020; Simonsen et al., 2021).

Complex terrain and drastic changes in elevation could bring extra uncertainty in the merged time series. The beam-limited footprint of a radar altimeter with radius up to kilometres makes it difficult for the radar altimeter to accurately measure ice surface height in those areas. Terrain undulations on kilometre-scale or smaller might make the biquadratic surface polynomial approximate the local ice surface topography inaccurately, and thereby introduce errors into the correction for existing topography-induced height differences between the individual shots. Surface elevation observations from data products have

been relocated by the point of closest approach were used in this study to suppress the influence related to the excessive size of footprint. The possible terrain correction errors caused by small-scale relief can only be expected to be suppressed by the mean estimator. Thus, the uncertainties of average SEC rates (Figs. 4 and 5) for marginal areas with complex terrain are larger than those for the central ice sheet. It is also reflected in the estimation of intermission bias and ascending–descending bias (Frappart et al., 2016; Zhang et al., 2020), although we have used large amounts of data to fine-tune them for each grid cell,

which has been proven to ensure better self-consistency and reliability of the combined elevation time series (Zhang et al., 2020).

Additionally, interpolation or extrapolation of unobserved cells might also introduce uncertainty into the merged results, especially in steep and very active areas at the margins of the GrIS. The limited number of valid elevations, along with the greater uncertainty of several of them, would inevitably cause interpolation (extrapolation) error. This study used the EOF

reconstruction method to reduce the error, which can incorporate more temporal and spatial information to constrain the interpolation results. However, some interpolation with large uncertainty still exists in some steep or narrow glaciers at the margins of the GrIS. The first three outliers of the volumetric time series shown in Fig. 9 (c) are caused by this error. To avoid the large uncertainty caused by interpolation, (Sørensen et al., 2018) arbitrarily excluded all grid cells located on slopes exceeding 1.5˚, and (Schröder et al., 2019) exclude all data prior to 1992-04-14 from ERS-1, while we provided the merged

non-interpolated time series in the dataset.

Overall, the above factors might cause errors to our time series, but it is difficult to formally account for them. Thus, according to previous studies, a straightforward estimate of uncertainty was given in this study as described in Sect. 2.5. It is an empirical estimation, there may exist some underestimation due to the errors from above sources which are difficult to quantify.

## 5 Data availability

The surface elevation time series of the GrIS can be downloaded from the National Tibetan Plateau Data Center at http://dx.doi.org/10.11888/Glacio.tpdc.271658 (Zhang et al., 2021). In this repository, the time series is provided in NetCDF (.nc) format and named Surface_Elevation_Anomaly_Greenland_Monthly_5km_Grid.nc, which is easy to read or reanalyze with MATLAB and Python software. There are 9 variables in the .nc file, including longitude (lon), latitude (lat), time (time), SE anomaly before interpolation and its uncertainty (elev, elev_uncer), SE anomaly after interpolation and its uncertainty

(elev_interp, elev_uncer_interp), the drainage systems number (basin) and the flag of interpolation (flag_interp). The specific information of these variables has been indicated in the data file.

# 6 Conclusions

In this study, we developed a 30 year SE time series over the GrIS by combining ERS-1, ERS-2, Envisat, and CryoSat-2 satellite radar altimeter observations. A large number of operations, especially an updated plane-fitting least-squares regression strategy and an EOF reconstruction method, were performed to ensure that the time series has higher accuracy with monthly time resolution and $5 \times 5$ km spatial grid resolution. Validations with airborne laser altimetry observations and ESA GrIS CCI SECs indicated that our merged SE time series is reliable. In terms of the 5 year average SEC rates, the accuracy and dispersion of errors of our results were 19.3% and 8.9% higher than those of the CCI SECs, respectively. Benefiting from the finer correction of the inter-mission bias, the accuracy and dispersion of errors in our results were improved by up to 30.9% and 19.0%, respectively, in periods from 2006–2010 to 2010–2014.

The SECs and volume changes of the ice sheet are important variables that reflect the effects of climate change. As shown in Sect. 3, our data series can be used not only for studying long-term changes in the elevation and volume of the GrIS, but also for studying their temporal and spatial evolutions in detail on a temporal scale of up to 30 years. In particular, benefiting from the high temporal and spatial resolutions of our time series, the temporal and spatial evolution processes of ice loss from the main outflow glaciers in the GrIS can also be described in detail. These evolution processes are the response of the GrIS to oceanic and atmospheric changes driven by climate change. Thus, our merged time series provides an opportunity to examine the potential associations between ice sheet changes and climate forcing. The spatiotemporal patterns of accelerating or decelerating SEC of the GrIS, caused by shifts in atmospheric forcing and oceanic forcing driven by NAO phase transformation, reveal the sensitivity of the GrIS to climate forcing.

The mass balance of an ice sheet is a climate-related variable that has greater scientific value than elevation change. If combined with an appropriate ice density model, we could obtain a mass balance time series from our merged time series with much higher spatial resolution and longer temporal coverage than that of either GRACE or GRACE-FO. This could have advantages for studying mass change in small basins, especially the mass balance of outflow glaciers, thereby improving the estimation accuracy of the mass balance of the GrIS and reducing the uncertainty of projections of future sea level change.

**Author Contributions.**

BZ performed the calculation and wrote the manuscript. ZW contributed to the conception of the study. JA advised on validation and revised the manuscript. TL supervised the work. HG contributed to discussions and analysis of the results. All authors contributed to improvement of the manuscript.

**Competing interests.**

The authors declare that they have no conflict of interest.

**Acknowledgements.**

We would like to thank the organizations that shared their datasets and software for use in this study. The ERS-1, ERS-2, Envisat, and CryoSat-2 observations and the GrIS CCI SECs were provided by the European Space Agency and the airborne elevation data were provided by the National Snow and Ice Data Center. All geographical plots were produced using Generic Mapping Tools. We thank James Buxton MSc, from Liwen Bianji (Edanz) (www.liwenbianji.cn), for editing the English text of a draft of this manuscript. Topical editor Tao Che and three anonymous reviewers are thanked for their comments that helped clarify and improve the manuscript.

**Financial support.**

This work was supported by the National Key Research and Development Program of China (2018YFC1406102), National Natural Science Foundation of China (42006184, 41941010), and Strategic Priority Research Program of the Chinese Academy of Sciences (Grant No. XDA19070100).

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

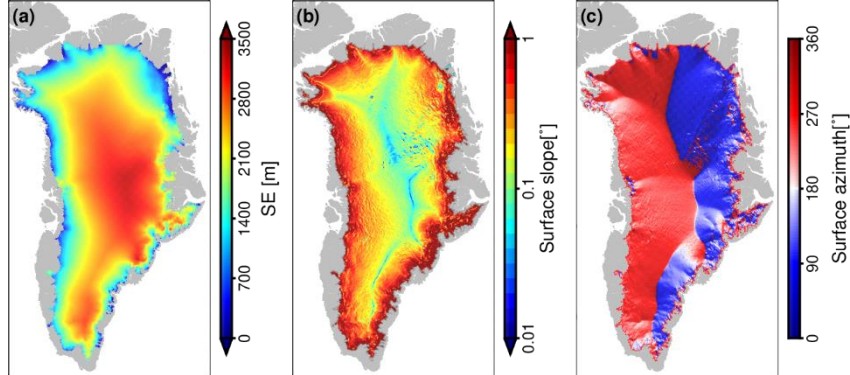

**Figure 1:** (a) SE of the GrIS DEM at reference time 2010.0, and (b) surface slope and (c) azimuth derived from the DEM.

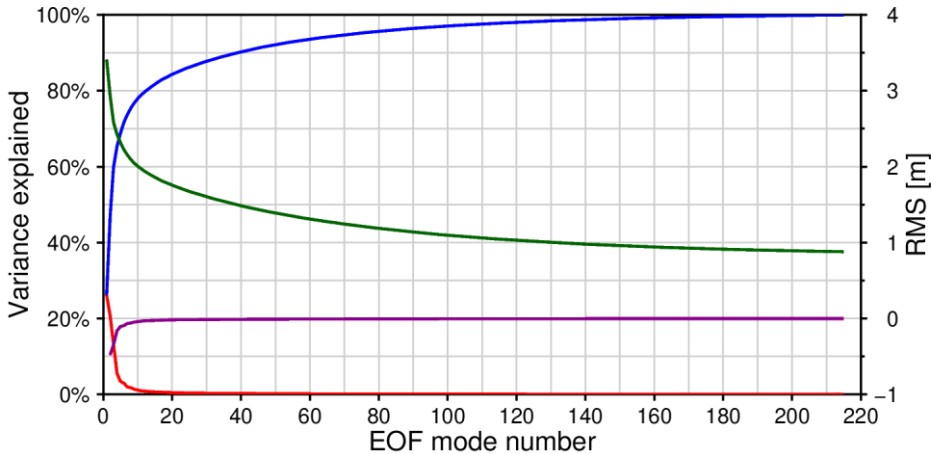

**Figure 2:** Percentage of variance explained (red line) and cumulative variance explained (blue line) by each EOF mode; RMS error (green line) and its derivative (purple line) of the difference between the reconstructed time series of monthly elevation change and that of 600 observation in different EOFs.

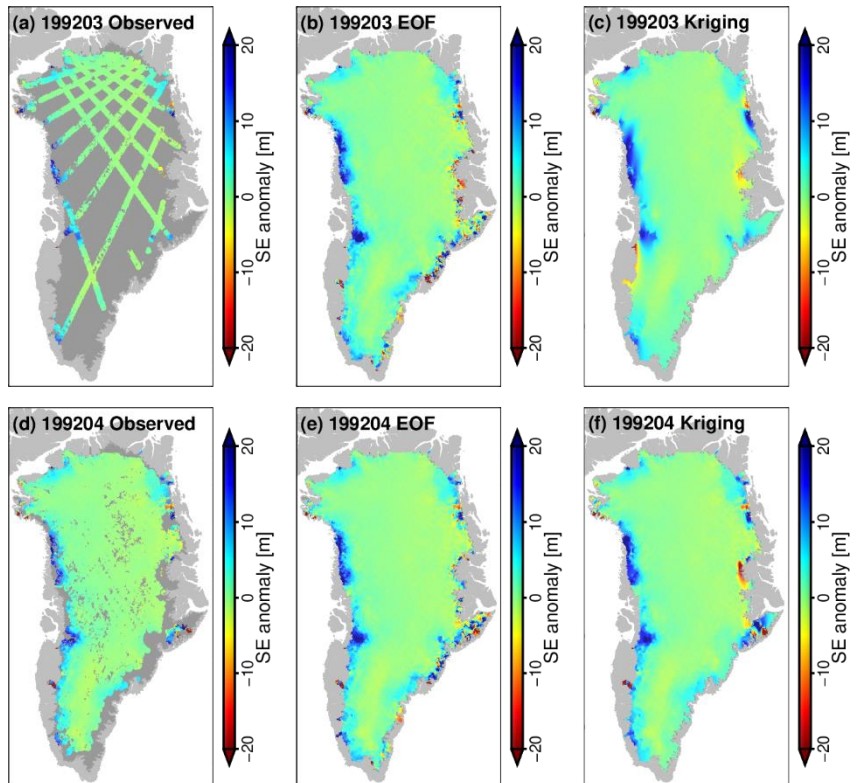

**Figure 3:** Interpolation performance of EOF reconstruction and ordinary kriging: (a), (b), and (c) are the results for March 1992 observation, EOF reconstruction, and ordinary kriging interpolation, respectively, and (d), (e), and (f) are the same for April 1992.

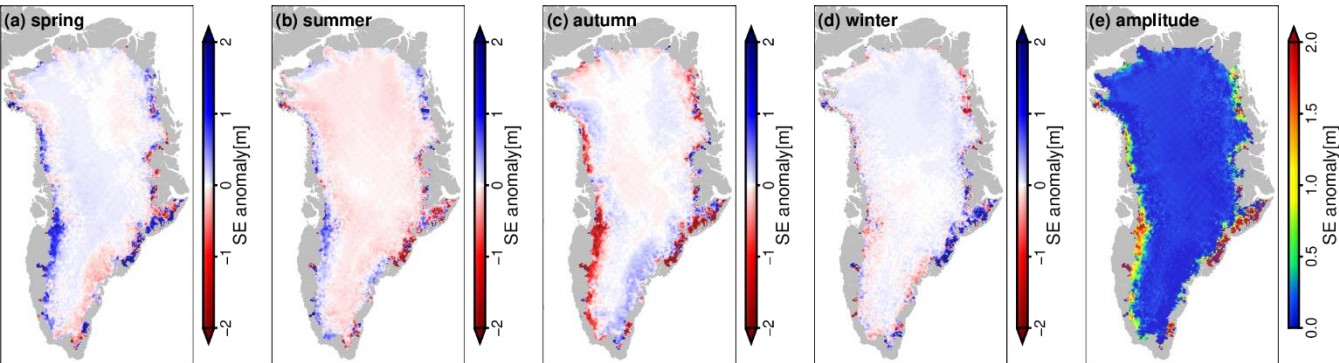

**Figure 4:** Climatological maps of SE anomaly averaging season by season: (a) spring (January-February-March), (b) summer (April-May-June), (c) autumn (July-August-September), and (d) winter (October-November-December) and (e) the amplitude of corresponding annual variation over the periods of 1991–2020 from the merged elevation time series.

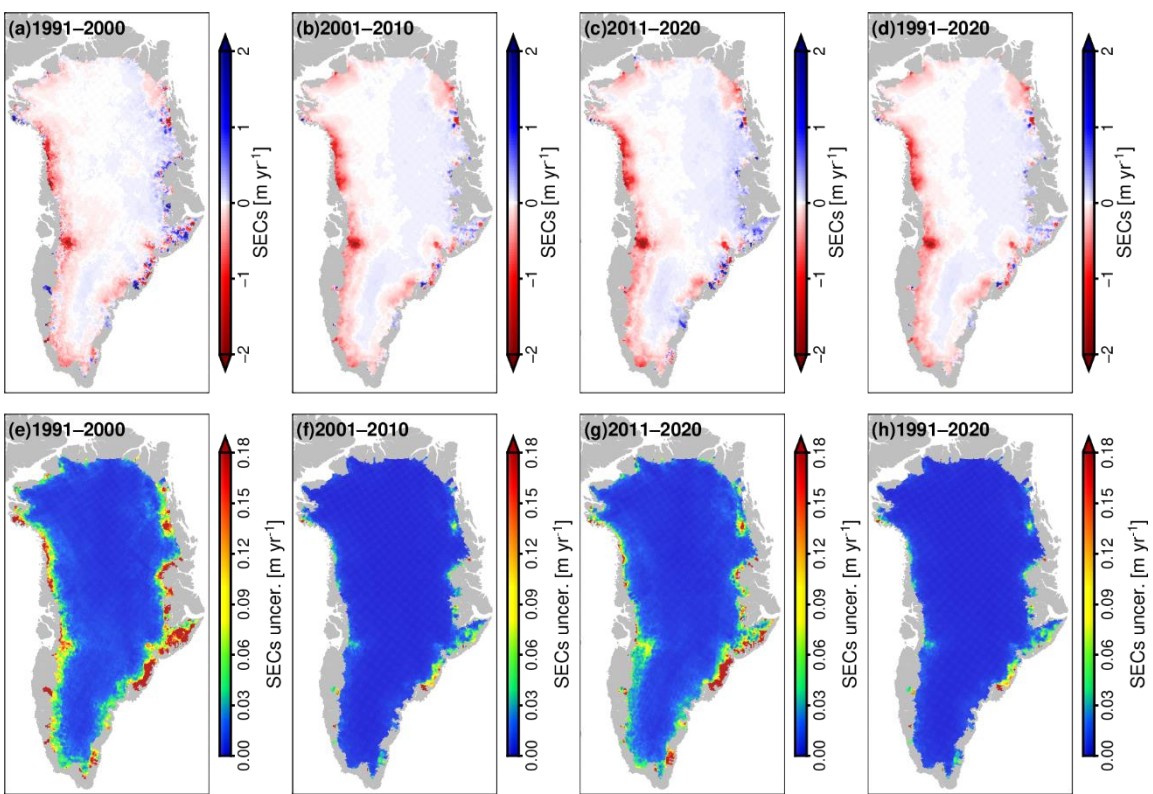

**Figure 5:** Maps of long-term SECs and their uncertainties from the combined elevation time series over the periods of (a) and (e) 1991–2000, (b) and (f) 2001–2010, (c) and (g) 2011–2020, and (d) and (h) 1991–2020.

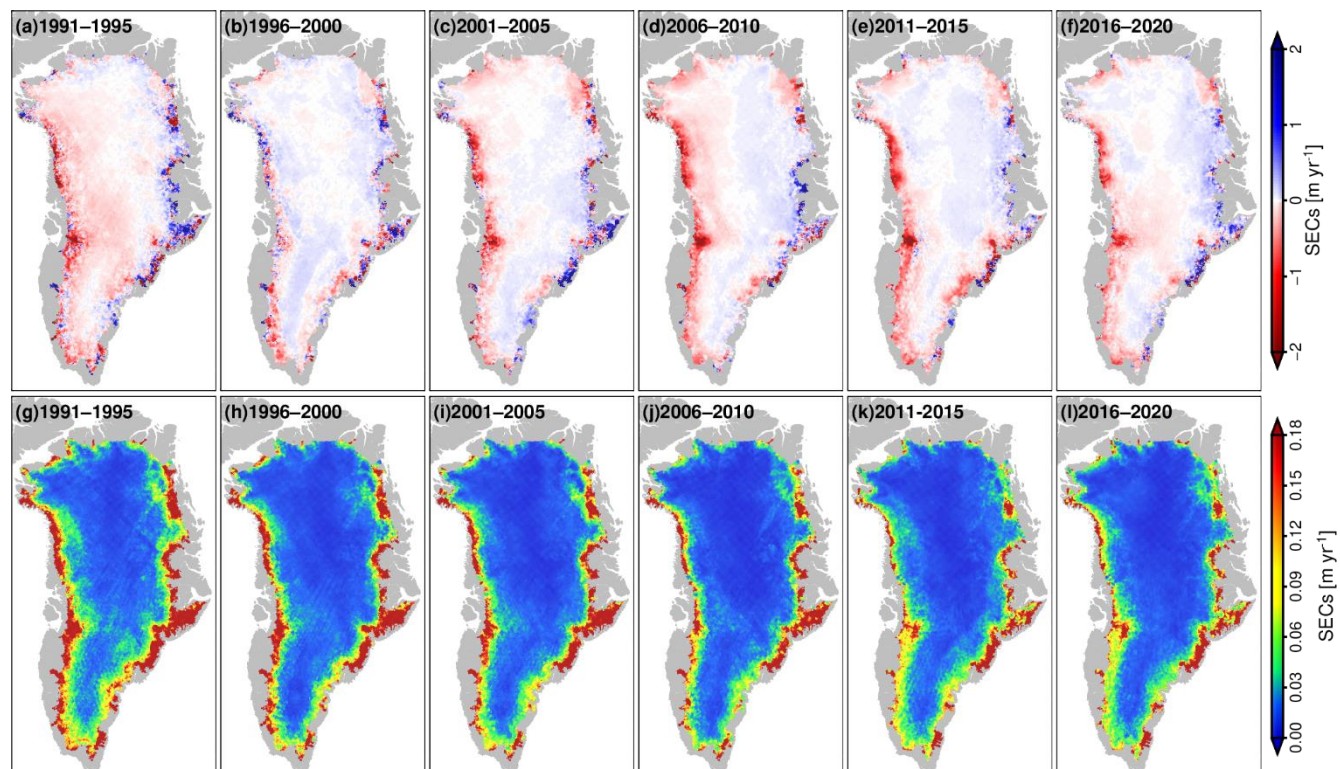

**Figure 6:** Maps of long-term SECs and their uncertainties from the combined elevation time series over the periods of (a) and (g), 1991–1995, (b) and (h) 1996–2000, (c) and (i) 2001–2005, (d) and (j) 2006–2010, (e) and (k) 2011–2015, and (f) and (l) 2016–2020.

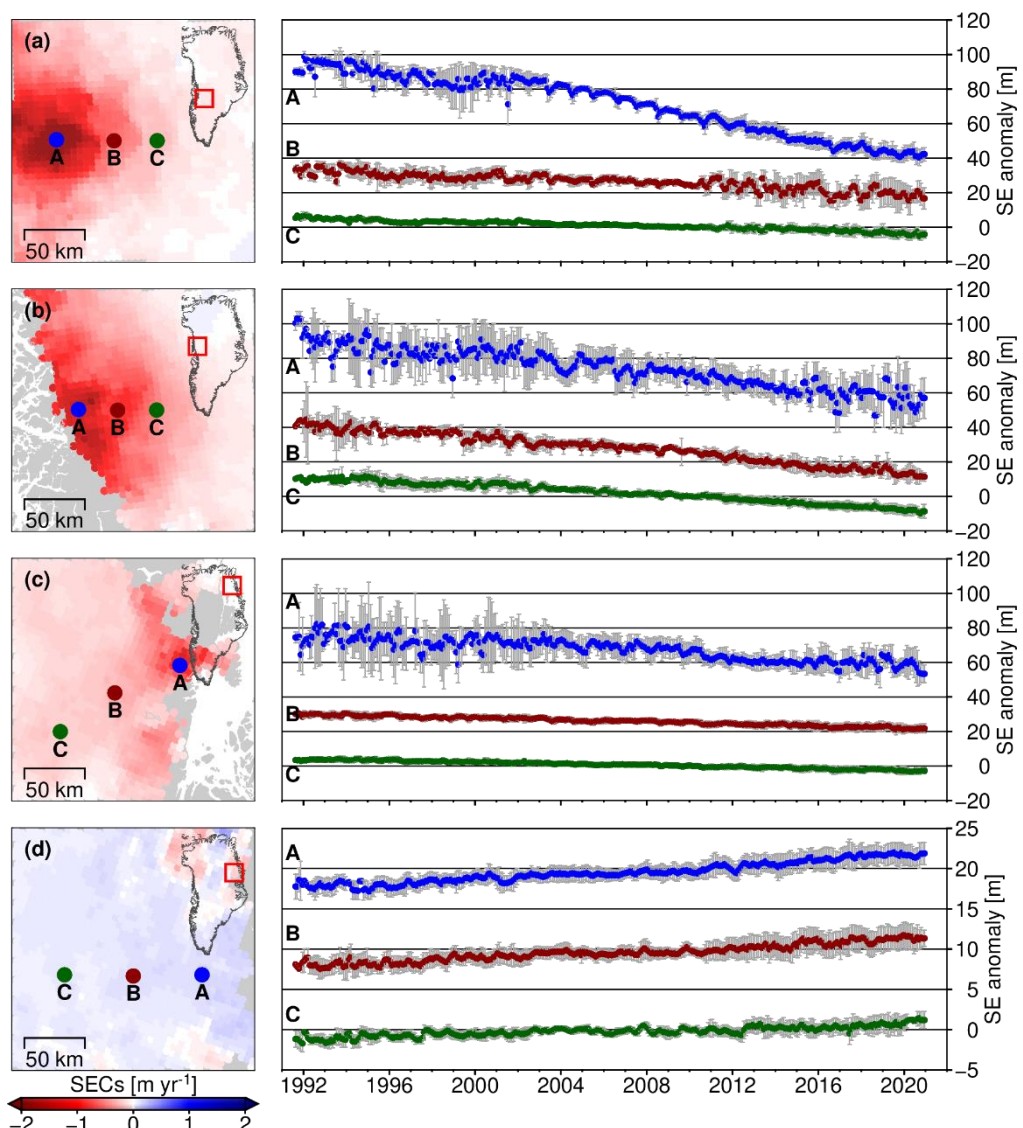

**Figure 7:** SE anomaly time series near (a) Jakobshavn Isbræ, (b) Upernavik Isstrøm, (c) Zachariæ Isstrøm and Nioghalvfjerdsfjorden glacier, and (d) King Christian X Land. The locations of the selected points (A, B, and C) are marked in the left-hand maps of elevation change over 1991–2020. The time series and the 1σ uncertainty ranges for each point are given in the right-hand plots with the time series of points A and B shifted along the SE anomaly axis for better visibility.

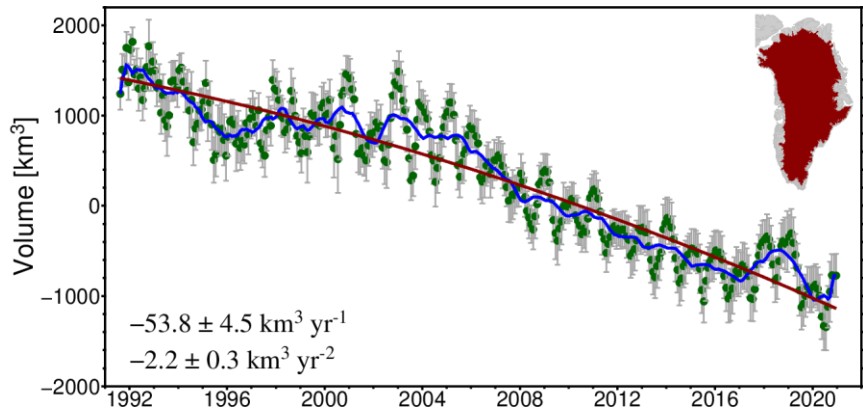

**Figure 8:** Volume change of the entire GrIS south of 81.5°N from our merged altimetric time series (green dots) and its corresponding result after removing seasonal oscillations using a 13 month moving average (blue solid curve). The solid red line is the best-fit quadratic curve for the linear and quadratic trend estimates of volume change. The grey error bars show the 1σ uncertainty range of the altimetry data. The red shaded area in the inset indicates the coverage of the GrIS with reference to the Greenland drainage system boundaries in Zwally et al. (2012).

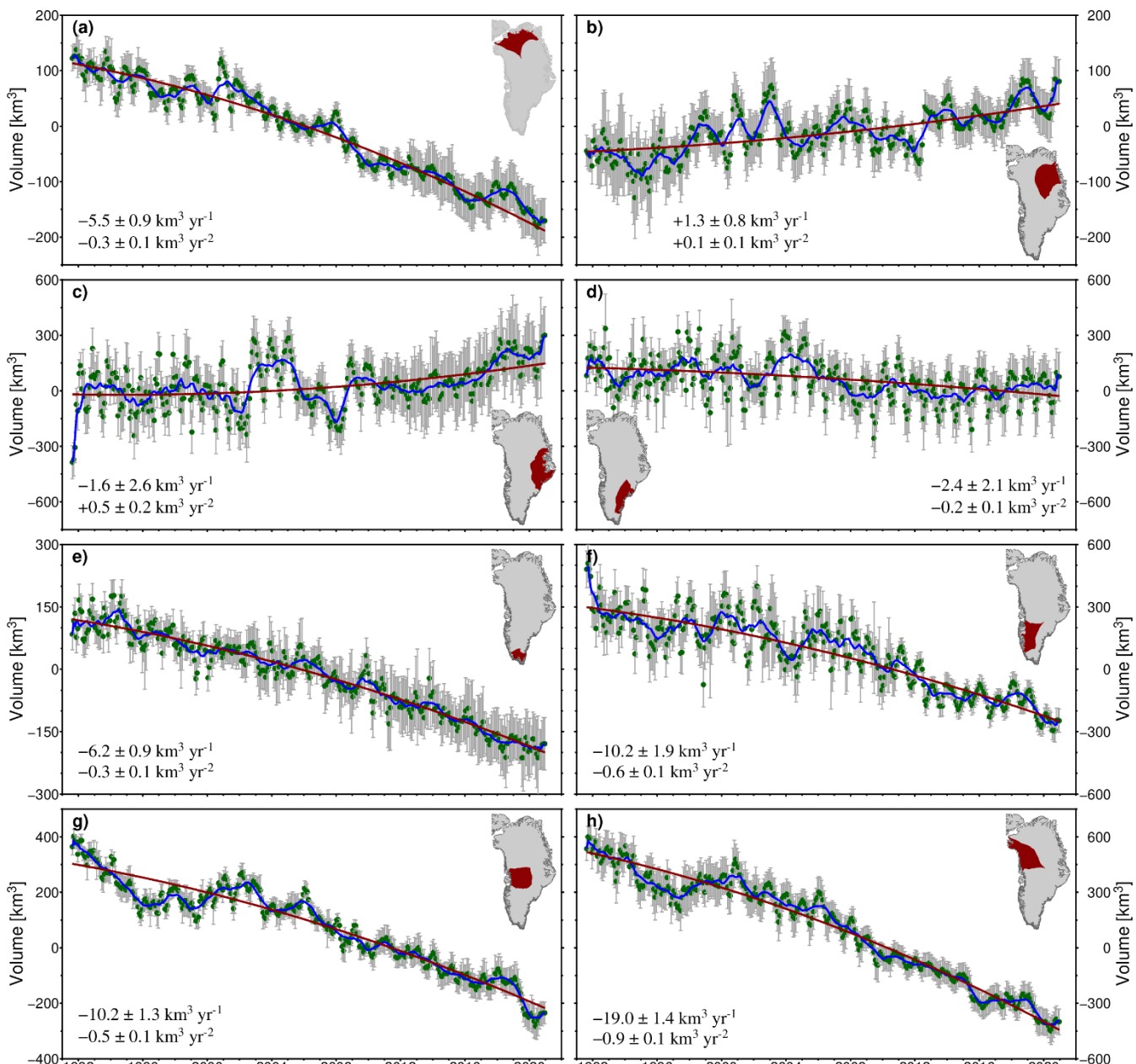

**Figure 9:** Volume change of sub-regions south of 81.5°N from our merged altimetric time series (green dots) and their corresponding results after removing seasonal oscillations using a 13 month moving average (blue solid curves). The solid red lines are the best-fit quadratic curves for the linear and quadratic trend estimates of volume change. The grey error bars show the 1σ uncertainty range of the altimetry data. The red shaded area in the inset of each panel indicates the coverage of the sub-region with reference to the Greenland drainage system boundaries in Zwally et al. (2012).

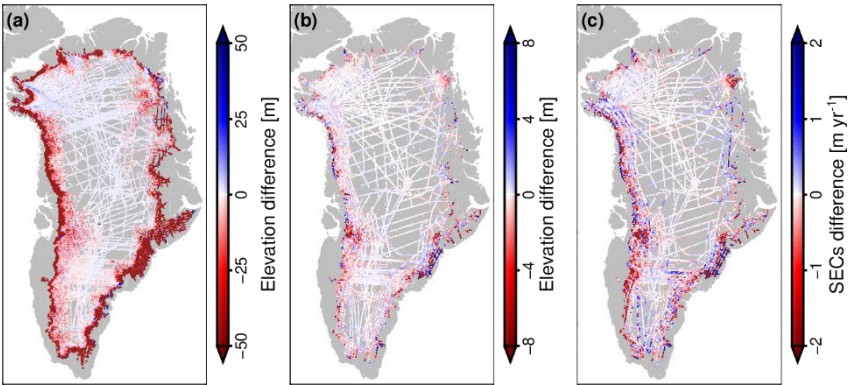

**Figure 10:** Validation with ATM laser altimeter observations: (a) differences between SE derived from ATM L2 and our merged time series; (b) differences between SE derived from ATM L4 and our merged time series; and (c) differences between SEC derived from ATM L4 and our merged time series .

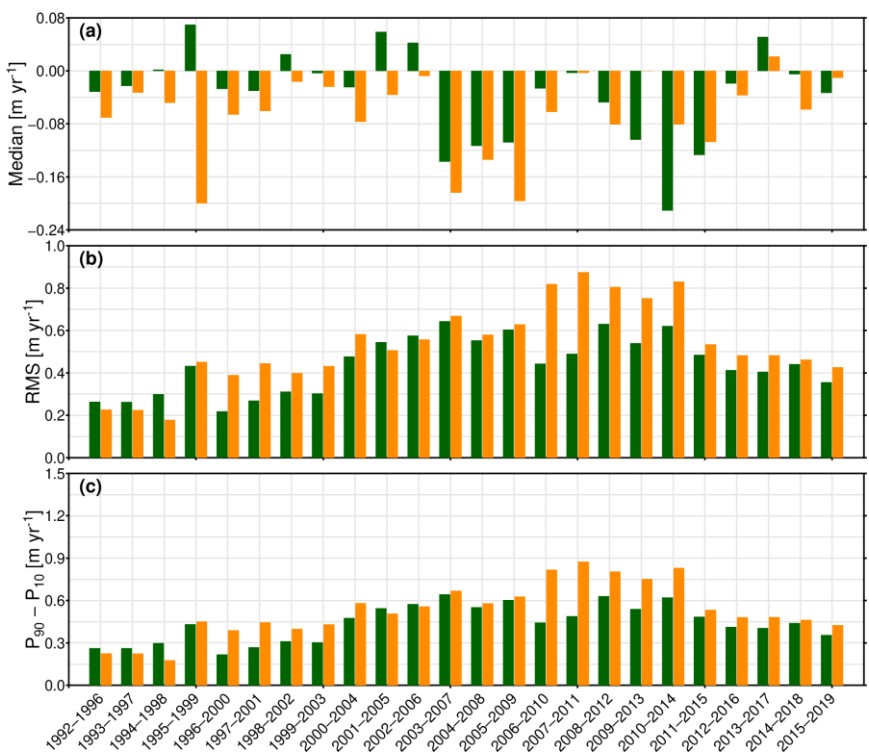

**Figure 11:** Validation with GrIS CCI SECs: (a) median, (b) RMS error, and (c) $P_{90}$-$P_{10}$ of the SEC differences between CCI (orange) and those derived from our meged time series (green) at 5 year intervals during 1992–2019.

**Table 1.** Statistics of the results of validation with ATM laser altimeter observations. The median, RMS error, and $P_{90}$–$P_{10}$ of the biases are given.

| Validation data | | Statistics of the comparison with validation data | | |
|---|---|---|---|---|
| Data | Variable | Median | RMS | $P_{90} - P_{10}$ |
| ATM L2 | SE (m) | -2.82 | 99.43 | 145.25 |
| ATM L4 | SE differences (m) | −0.02 | 8.84 | 3.78 |
| ATM L4 | SECs (m yr$^{-1}$) | −0.00 | 3.63 | 0.95 |

**Table 2.** Statistics of the results of validation with GrIS CCI SECs. The median, RMS error, and P90–P10 of the biases are given.

| | Statistics of the comparison with ATM L4 SECs | | | | | |
|---|---|---|---|---|---|---|
| | All Periods | | | Periods across the overlap of Envisat and CryoSat-2 | | |
| | Median (m yr$^{-1}$) | RMS (m yr$^{-1}$) | $P_{90} - P_{10}$ (m yr$^{-1}$) | Median (m yr$^{-1}$) | RMS (m yr$^{-1}$) | $P_{90} - P_{10}$ (m yr$^{-1}$) |
| GrIS CCI | -0.05 | 0.57 | 0.79 | -0.04 | 0.81 | 1.16 |
| This study | −0.03 | 0.46 | 0.72 | -0.09 | 0.56 | 0.94 |