# Peer review of "A 30 year monthly 5 km gridded surface elevation time series for the Greenland Ice Sheet from multiple satellite radar altimeters"

_Earth System Science Data, 2021_

## Referee Comment (RC2)

Zhang et al. presented a new dataset of the surface elevation changes over the Greenland Ice Sheet at monthly intervals spanning 30 years from 1991 to 2020. The long records build on careful corrections of the biases within and between four satellite radar altimeters based on plane-fitting. The authors used empirical orthogonal functions obtained from the 30-year observations (esp. the later, more accurate and complete measurements from Envisat and CryoSat-2) to fill in spatial gaps and reconstruct 5-km-resolution gridded fields.

Since this is a data-oriented journal, my comments below are mainly from a data user point of view. I also provide some technical and editorial comments at the end.

The data provided through the National Tibetan Plateau/Third Pole Environment Data Center are freely accessible through the link provided. The data file in NetCDF format can be easily read. I did run into a few issues listed below.

(1) The description of the field 'elev_ interp' is "interpolated surface elevation timeseries base on EOF Reconstruction and Kriging" and differs from what are shown in Figures 3b and 3e (and stated in the figure caption) as they are only from EOF reconstruction. Then I found it confusing as in the manuscript on line 160, where it is stated that "The missing values in the retained gridded time series are interpolated using ordinary kriging", implying that the final interpolated results are indeed based on both EOF reconstruction and kriging. If this is the case, is it still fair to compare your results with those only using kriging (as in Figure 3)?

(2) The field 'elev_ interp' seems to contain large outliers. Taking the one for February 1992 as an example, the minimum and maximum values of SE are -459 m and 558 m, versus -216 m to 34 m in the non-interpolated field. Similarly, larger positive and negative values are found along coastal regions as we can tell from comparing Figures 3b and 3e. These make me wonder how reliable are the monthly, interpolated fields. Even the authors chose not to use them when comparing with the CCI products (Line 339). Could the authors provide guidance for (i) the cautions users need to take when using the monthly, interpolated fields and (ii) handling outliers in these fields?

(3) Related to the reliability, the uncertainties provided are definitely helpful. Yet, as they were approximated by deviations from a regular trend + acceleration + seasonal time series (eq. 9), is it possible that these uncertainties, esp. the ones for interpolated SE, are underestimated?

(4) The description and names of many variables in the NetCDF file contain spelling errors, e.g., 'longitudt' should be longitude; 'latitudt' should be latitude; 'degress_north' should be 'degrees_north'; 'Baisn Num' should be 'Basin Num' and is based on Zwally et al., 2012 Antarctic and Greenland Drainage Systems. None of these affect the use of the data, but there are too many typos in the meta data.

Methodology description needs to be improved.
Section 2.3 largely builds on the authors' previous work on the Antarctica Ice Sheet and published in Remote Sensing. Yet, I found this subsection difficult to read, largely due to the repeated use of same or very similar letter symbols in eqs. 1-5 but they actually have different meanings. Some terms are not introduced at all, such as $(-1)^{AD}$ in eq. 1 and $(-1)^{im}$ in eq. 2. This section needs to be rewritten to improve its clarity.

Section 2.4 and result parts related to EOF: this is relatively new in this work and needs further analyses and discussion in terms of the validity of assumptions, quality of the interpolated results. For instance, the assumption behind EOF is that the spatial patterns of elevation changes "are stationary in time" (Line 53), which do not hold as evident in Figures 4 and 5 and numerous studies. How would temporal variations of spatial patterns, esp. those during the ERS period and the later Envisat-CryoSat-2 period, affect the EOF reconstruction?

It is not clear to me what is the authors' basis for claiming "the superiority" of EOF over kriging (Line 73-74). My concern, as raised above, is unreliable EOF reconstruction and the large values along the coast.

In the volumetric time series (Figures 6-8), it would be helpful to add the ones based on the CCI products with associated uncertainties as another cross-validation.

Figure 8c shows a sharp volumetric increase at the beginning of the time series (more than 300 km3 from 1992 to 1993). This doesn't seem to be correct. Could you double-check or validate with independent data?

Minor comments:
It is important to mention EOF in the abstract.
Line 48: specify what kind of data
Line 57: 'Therefore …'; I don't see a causal link between the limitations of the other approaches and the solution to be offered by data combination. Or is something missing in this last sentence of the paragraph?
Line 93-95 fit better in methodology.
Line 208: change 'derived from' to 'caused by'
Line 227: 'Greenaldn' should be 'Greenland
Line 298: what is 'official relocation'?

---

## Referee Comment (RC3)

**Review of Zhang et al. (2021)**

General comments:

This study describes the methods used to produce a new dataset of surface elevation changes of the Greenland Ice Sheet from satellite radar altimetry observations acquired by ERS-1, ERS-2, Envisat and CryoSat-2 spanning the period 1991 to 2020 and features some illustrations of this dataset over specific glaciers of the Greenland Ice Sheet. The main novelty in the methods described here is the use of EOF modes to fill in the monthly grids of surface elevation change.

I have some important general comments and questions that need to be addressed before I can recommend this manuscript for publication:
- Is it a reasonable assumption to assume that 'spatial patterns of GrIS SECs are stationary in time' (L153) in your EOF reconstruction? The pattern of SECs has evolved from the 1990s to the 2003-2020 period. The authors themselves show this in Figure 4 where we can see the spread of thinning further inland between panels a (1991-2000) and c (2011-2020). How robust is this assumption?
- The SECs dataset comes with an uncertainty estimation, which is great but I am not convinced by the uncertainty assessment made here. I understand that using the MAD is straightforward but I wonder if it is a good measure of the uncertainty of the measurement. While I recognise that it is difficult to formally account for the different sources of uncertainty in the altimetry SECs measurements, some useful information/discussion could be added regarding the uncertainty assessment – for instance a discussion of which step in your processing contributes to larger uncertainties (is it the inter-mission calibration, the formulation-n of the plane-fit model…?)
- I would rename Section 4 'Comparison to Independent Datasets' as this section is not really an uncertainty assessment but more a validation/comparison analysis to the ATM and CCI datasets. I suggest to move the first subsection 'Error sources' at the end of this section and rename it 'Limitations of the dataset' or something along those lines.
- In the 'Results' section on the analysis surface elevation anomaly time-series, the authors look at decadal trends in SECs over specific glaciers which highlight the long record that they have produced. This is a good illustration of the potential use of the dataset and it would be good to also feature an illustration of the ability of the dataset to look at seasonal changes in elevation change to highlight the high temporal resolution of their dataset, or at least to discuss this in the text.
- Do you see any step change in the SEC time-series following the 2012 extreme melt event in the interior of the ice sheet? Could you comment on whether this artefact is present in your time-series or if your processing scheme is able to correct for this effectively?
- It is straightforward to download the data from the link provided but there are quite a lot of typos in the metadata of the NetCDF file: 'latitudt', 'longitudt' or for instance the description of the basins variable states: 'Antarctic_Drainage_System_Boundaries_and_Masks' when it should be 'Greenland_Drainage_System_Boundaries_and_Masks'.

I made some specific comments and suggestions below, which I hope will help improve this paper.

Specific comments:

**L8:** 'for study of ice sheet variation and its response to climate change', please reformulate

**L26:** You also need a density model for the snow and firn layer in addition to a model of the distribution of the ice layers within the firn column to convert volume change to mass change. Please clarify this sentence.

**L28:** Please be more specific in this sentence, a long-term time series of EC is essential to assess the impact of climate change on the ice sheets rather than 'to assess climate change directly'.

**L56:** I recall that the deconvolution method from Slater et al (2019) does provide time-variable penetration depth over the interior of the Greenland Ice Sheet.

**L57-58:** 'a more reasonable approach' than? I would argue that the reason to combine data from several radar altimetry missions is that it is the only way to get a long record of surface elevation changes. I would remove this sentence, as it doesn't link with what was said in the paragraph.

**L59:** By irregular, do you mean that the satellite tracks deviate from the ground-tracks? I think you need to be a bit more specific here as this sentence could be misread. ERS-1/2 and ENVISAT have a repeat cycle and CryoSat-2 a drifting orbit but all missions have regular a ground track pattern.

**L64:** 'for to unobserved grid cells'

**L85:** Specify how similar the sensors are (for instance they all use Ku-band etc). You could also mention Sentinel-3 here. Could your processing scheme be applied to Sentinel-3 data as well?

**L93:** typo 'SARIn'

**L171-172:** How do you estimate the seasonal signals here? Do you use the terms from the plane fit model or do you estimate the seasonal components using a time-series decomposition technique?

**L175:** Can you add the average proportion of the monthly grid that has be filled in using the EOF modes for each mission? Data are rather sparse during the ERS-1/2 periods compared to the CryoSat-2 period when only small gaps between tracks occur. It would be useful to add a sentence in the text to reflect this.

**L204:** Did you look at trends in surface mass balance to support this claim?

**L217:** Are there limitations of your dataset to look at SECs over some small glaciers? Is there an optimal area size for glaciers for which your dataset would be the most useful? Is your dataset reliable close to the termini of glaciers where altimetry measurements are usually sparse and the slope is high?

**L227:** typo 'Greenland Ice Stream'

**L235-236:** Not all fluctuations in SECs are caused by climate change, in case of short-term fluctuations it is hard to distinguish between natural climate variability and climate change

**L292:** Here the advantage is to use radar altimetry instead of laser altimetry to look at elevation changes induced by SMB processes. I would add in the sentence 'by combining data from multiple radar altimetry missions'

**L553:** typo 'left-hand maps'

**Fig 9:** Please clarify the caption, it's unclear from the caption what is shown on Figure 9b. Reading the caption alone, it looks like maps a and b are showing the same thing. Also, what do you by the same epochs for map c?

**Table 2:** What do you mean by 'periods across the Envisat and CryoSat-2 connections?' I would maybe say 'overlap' rather than 'connections'.

---

## Author Comment (AC1)

Dear the reviewers:

First of all, we would like to take this opportunity to thank the reviewer for your constructive comments and relevant questions. By adding the answers/revisions to these questions to the revised version of the manuscript, we feel that the quality of the manuscript has been improved. A revised manuscript has been submitted, and all of corrections/modifications are only included in the revised manuscript for the sake of non-repeat. Extra answers to your concerns and questions are presented as follows.

**Reviewer 1:**

**Comments**

Review of manuscript "A 30 year monthly 5 km gridded surface elevation time series for the Greenland Ice Sheet from multiple satellite radar altimeters" submitted to Earth System Science Data (https://doi.org/10.5194/essd-2021-293)

The manuscript uses a long record of satellite altimeter from ERS-1, ERS-2, Envisat and Cryosat-2 to construct a database of surface elevation change (SEC) of Greenland ice sheet at monthly temporal resolution and 5x5 km spatial grid resolution. The authors used an updated strategy plane-fit method to cross-calibrate data from the different missions, a large dataset to correct for the intermission bias and a method to suppress the interpolation error. Results are validated using NASA's IceBridge airborne data over Greenland and ESA's Climate Change Initiative.

 Given the ongoing impact of global warming on Arctic environment, the subject is timely and the output dataset is certainly needed. The advantage of this dataset resides in its longest record so far (30 years), making use of a series of multiple satellite altimetry missions. In addition to validation against external data, the authors claim the reliability of the dataset based on the methods used to eliminate system biases between the altimeter systems. This seems logical but I have a concern about the validation. What is being offered is just a comparison against the external data (IceBridge), which still have their sources of error although laser altimeter is known to be more accurate than radar altimeter. Cross comparison is always good but it does not necessarily mean validation.

Data on regions that experience thinning and thickening of the elevation are presented with explanation related to meteorological and environmental factors and links to previously established data. This is useful information. Similar useful information is presented on regional volume change in relation to NAO and this is supported by findings from previous studies.

Overall, I think the authors are presenting a credible study and the database adds to the growing number of studies on surface elevation, and eventually mass balance, of Greenland ice sheet using the existing longest altimeter data record so far.

**General comments:**

Nothing is mentioned about elevation change from ICESat. A couple of sentences in the Introduction and comparison of results would be useful (if results from ICESat exists), e.g., is thinning along the west coast – is it also confirmed from ICESat?

Answer: Thank you for the suggestion. We have added in Sect.3.1 in the revised manuscript.

While there are comments on thinning of the ice sheet in the west coast, no comments are offered about thickening of the ice sheet in the east coast, particularly south of 73 deg. latitude. (Figure 6 shows much less fluctuations in the King Christian X Land)

Answer: Thank you for the comment. In the revised manuscript, we have given some comments on thickening of the ice sheet in the King Christian X Land in the east coast. There are two reasons why we only took the King Christian X Land as the representative area for discussion. The one is that the King Christian X Land was selected according to the long-term trend from 1991 to 2020 (Fig. 4 (d)). It is a typical thickening area, with an altitude of more than 2000m (see Fig. 1(a)), which is usually above the mass balance line of Greenland Ice Sheet. Its signals of thickening mainly reflect the increase in SMB derived from a combination of high snowfall and low surface melting (Simonsen et al., 2021; Wood et al., 2021). The other reason is that elevation observations from radar altimeter is less affected by steep terrain than those in the east coast in south of 73 deg. Latitude (see Fig. 1(b), and Fig.1 of Sørensen et al. (2018)). Sørensen et al. (2018) discarded these grid cells with slopes greater than 1.5 deg. in the east coast in south of 73 deg., because the large uncertainty come from the steep terrain. The aim of this study is to provide a dataset. Thus, we kept the data as much as possible in case of other studies need.

As the title asserts, data are available on monthly basis. However, data are presented in all graphs on yearly basis. It would be nice to present the change in elevation in summer compared to winter somewhere in the manuscript, or at least comment on this aspect.

Answer: Thank you for the suggestion. We have made relevant revision in the revised manuscript.

Only hints on the method of calculation of volume change are mentioned in the Introduction and Section 2.6. A few lines to describe the calculations can be inserted at the end of Section 2.3 or in a dedicated section.

Answer: Thank you for the suggestion. We have added the describe of calculation of volume change in at the end of the Section 2.4 in the revised manuscript. It is added at the end of Section 2.4 instead of Section 2.3 because Section 2.4 is also used to calculate the elevation time series. The volume change shown in this study is calculated from the result of Section 2.4.

Why is the difference between ATM data and the present data large in the peripheral areas of Greenland, where the SE is low? Larger uncertainty in the derived elevation?

Answer: The large differences between ATM data and the present data in the peripheral areas of Greenland is because that the radar altimeter has larger measurement errors in steeper and active areas due to the higher surface slope and roughness because of the its excessive footprint size. It has been discussed in some previous studies, such as Simonsen and Sorensen (2017), Schröder et al. (2019) and Zhang et al. (2020). The elevations accuracy of radar altimetry over the flat interior of the ice sheet is indeed higher than that in the peripheral areas of the ice sheet.

In section 4.1, error sources are described. Can you attach numerical figures of uncertainty due to each source?

Answer: Thanks for pointing out this issue. But unfortunately, as the Anonymous Referee #3 said that "it is difficult to formally account for the different sources of uncertainty in the altimetry SECs measurements". It is difficult for us to obtain the numerical uncertainty caused by the signal penetration, excessive footprint size and interpolation and extrapolation. On the one hand, we do not have enough high-accuracy surface observations to estimate them. On the other hand, we still have some deficiencies in understanding the mechanism of their errors. Therefore, we cannot attach numerical figures of uncertainty due to each source. For this reason, we have renamed 'Error sources' to 'Limitations of the dataset' in the in the revised manuscript following the comments of the Anonymous Referee #3.

I think there is inconsistency between using SEC and SE. Please check.

Answer: Thanks for pointing out this issue. In this study SE is short for surface elevation; SEC is short for surface elevation change. The different between them is "change". Thus, SE is often associated with time series, while SEC is often associated with rates. But sometimes they are also associated with differences, meaning the differences between different SEs (SECs). We have checked them in the revised manuscript.

I could not determine the correctness of equations 1-5.

Answer: We have checked them again, and believe that they are correct. Eq. (1)–(3) are based on our previous study (Zhang et al., 2020), which is based on many previous studies, such as Hwang et al. (2016), Yang et al. (2019), Flament and Remy (2012), Mcmillan et al. (2014), Mcmillan et al. (2016), and Simonsen and Sorensen (2017). It can be said that Eq. (1)–(3) are the result of the synthesis of these studies. Eq. (4) is the deformation of Eq (3). Eq. (5) is a natural calculation of the mean. The similar derivation of Eq. (4)–Eq. (5) can also be found in Zhang et al. (2020) and Hwang et al. (2016). In addition, "∇" in equation 4 should be "Δ". We have corrected it in the revised manuscript.

Not sure what you mean by "surface aspect" in Fig. 1.

Answer: Thanks for pointing out this issue. Here, "surface aspect" means the direction of surface slope. Referring to Zwally et al. (2012), we find that "surface azimuth" is better than "surface aspect". We have changed it to "surface azimuth" in the revised manuscript.

**Minor specific comments**

Abstract:

The sentence "The accuracy and reliability of the time series is reliable". Then the following sentence has more quantitative data in the comparison. There is no need for the first sentence then.

Answer: Thank you for the suggestion. We have removed it in the revised manuscript.

Introduction:

Line 24: does GrIS really contribute to the sea level rise 1.4 times more than the Antarctic ice sheet? We know that melting GrIS has increase the sea level by about 1 cm in the past 3 decades. It would be better to quote actual numbers of sea level rise caused by Greenland and the Antarctic melting instead of just quoting the ratio.

Answer: Thanks for pointing out this issue, and we have made relevant revisions in the revised manuscript. From Shepherd et al. (2020), the average sea level contribution of the Greenland Ice Sheet is $0.42 \pm 0.04$ mm yr$^{-1}$ over the period from 1992–2018 (26 years). And from Shepherd et al. (2018), the total contribution of the Antarctic Ice Sheet to mean sea level is $7.6 \pm 3.9$ mm over the period from 1992–2017 (25 years), about 0.30 mm yr$^{-1}$. it is indeed about 1.4 times (0.42/0.30=1.4).

Line 90: ICE-1 re-tracker for ERS-1, ERS-2, OCOG re-tracker for Cryosat-2 …. Etc. It would useful if the authors add references to those methods.

Answer: Thanks for pointing out this issue, and we have added references in the revised manuscript.

Results:

Line 249: "Six of the eight drainage system ....". Can you show those drainage areas on the map?

Answer: Thanks for pointing out this issue, those have been shown in Fig 8. And we have added specific figure references in the revised manuscript.

Line 282: "... have been given in the figures mentioned here". Which figures?

Answer: Thanks for pointing out this issue, and we have rewritten it in the revised manuscript. The figures are the Figs 4–8.

**References cited in authors' response:**

Aschwanden, A., Fahnestock, M. A., and Truffer, M.: Complex Greenland outlet glacier flow captured, Nature Communications, 7, 10524, doi:10.1038/ncomms10524, 2016.

Bergmann, I., Ramillien, G., and Frappart, F.: Climate-driven interannual ice mass evolution in Greenland, Global Planet Change, 82-83, 1-11, https://doi.org/10.1016/j.gloplacha.2011.11.005, 2012.

Chambers, D. P., Mehlhaff, C. A., Urban, T. J., Fujii, D., and Nerem, R. S.: Low-frequency variations in global mean sea level: 1950–2000, Journal of Geophysical Research: Oceans, 107, 1-1-1-10, doi:10.1029/2001JC001089, 2002.

Church, J. A., White, N. J., Coleman, R., Lambeck, K., and Mitrovica, J. X.: Estimates of the Regional Distribution of Sea Level Rise over the 1950–2000 Period, J Climate, 17, 2609-2625, 10.1175/1520-0442(2004)017<2609:EOTRDO>2.0.CO;2, 2004.

Flament, T. and Remy, F.: Dynamic thinning of Antarctic glaciers from along-track repeat radar altimetry, J Glaciol, 58, 830–840, 10.3189/2012JoG11J118, 2012.

Hwang, C., Yang, Y. D., Kao, R., Han, J. C., Shum, C. K., Galloway, D. L., Sneed, M., Hung, W. C., Cheng, Y. S., and Li, F.: Time-varying land subsidence detected by radar altimetry: California, Taiwan and north China, Sci Rep-Uk, 6, 1–12, 10.1038/srep28160, 2016.

[revised manuscript text omitted]

Zwally , J. H. and Jun, L.: Seasonal and interannual variations of firn densification and ice-sheet surface elevation at the Greenland summit, J Glaciol, 48, 199-207, 10.3189/172756502781831403, 2002.

**Reviewer 2:**

**Comments**

Zhang et al. presented a new dataset of the surface elevation changes over the Greenland Ice Sheet at monthly intervals spanning 30 years from 1991 to 2020. The long records build on careful corrections of the biases within and between four satellite radar altimeters based on plane-fitting. The authors used empirical orthogonal functions obtained from the 30-year observations (esp. the later, more accurate and complete measurements from Envisat and CryoSat-2) to fill in spatial gaps and reconstruct 5-km-resolution gridded fields.

Since this is a data-oriented journal, my comments below are mainly from a data user point of view. I also provide some technical and editorial comments at the end.

The data provided through the National Tibetan Plateau/Third Pole Environment Data Center are freely accessible through the link provided. The data file in NetCDF format can be easily read. I did run into a few issues listed below.

(1) The description of the field 'elev_ interp' is "interpolated surface elevation timeseries base on EOF Reconstruction and Kriging" and differs from what are shown in Figures 3b and 3e (and stated in the figure caption) as they are only from EOF reconstruction. Then I found it confusing as in the manuscript on line 160, where it is stated that "The missing values in the retained gridded time series are interpolated using ordinary kriging", implying that the final interpolated results are indeed based on both EOF reconstruction and kriging. If this is the case, is it still fair to compare your results with those only using kriging (as in Figure 3)?

Answer: Thank you for the comment. In this study, we first interpolated the missing values in the gridded time series during 2003–2020 using ordinary kriging. Then, this gridded time series was used to solve EOF modes. These modes were then used to interpolate the missing values in the gridded time series during 1991-2002 based on EOF reconstruction. The purpose of solving EOF modes is to supplement the sparse monthly gridded data attributable to poor observations during ERS-1 and ERS-2 missions (during 1991-2002). So, we said "interpolated surface elevation timeseries (during 1991-2002 and 2003–2020) base on EOF Reconstruction and Kriging".  Fig.3 was used to show the performance of EOF reconstruction and ordinary kriging. Fig.3 (b) and (e) were interpolated based on EOF reconstruction; Fig. 3(c) and (f) were interpolated using ordinary kriging. We think it fair to compare their performance.

Maybe the word "retained" confused you. We have revised it in the revised manuscript.

(2) The field 'elev_interp' seems to contain large outliers. Taking the one for February 1992 as an example, the minimum and maximum values of SE are -459 m and 558 m, versus -216 m to 34 m in the non-interpolated field. Similarly, larger positive and negative values are found along coastal regions as we can tell from comparing Figures 3b and 3e. These make me wonder how reliable are the monthly, interpolated fields. Even the authors chose not to use them when comparing with the CCI products (Line 339). Could the authors provide guidance for (i) the cautions users need to take when using the monthly, interpolated fields and (ii) handling outliers in these fields?

Answer: Thank you for the comment. There might be several outliers along coastal regions. Most of the them come from extrapolation error. Limited by the natural defects of radar satellite

altimeter, it is difficult to obtain sufficient accurate measurements in areas with complex terrain and drastic changes in elevation, especially in the relatively narrow outlet glaciers along the ice sheet margin. In addition, even if we used $3\sigma$ outlier rejection criteria in least-squares regression and median filtering, some outliers will not be removed and thereby retained in the non-interpolated field. They are why there are still outliers while we used EOF reconstruction, which can incorporate more temporal and spatial information to constrain the interpolation results. To avoid the large uncertainty caused by interpolation, Sørensen et al. (2018) arbitrarily excluded all grid cells located on slopes exceeding 1.5°, and Schröder et al. (2019) exclude all data prior to 1992-04-14 from ERS-1. However, we retained them, mainly to retain some of the available observations in the dataset. It is also why we provided the merged non-interpolated time series in the dataset.

And we have added these discussions in Section 4.4 in the revised manuscript.

(3) Related to the reliability, the uncertainties provided are definitely helpful. Yet, as they were approximated by deviations from a regular trend + acceleration + seasonal time series (eq. 9), is it possible that these uncertainties, esp. the ones for interpolated SE, are underestimated?

Answer: Thank you for the comment. We agree it. The uncertainties provided in this study is a straightforward estimate according to previous studies. It is an empirical estimation, there may exist some underestimation due to the errors from other sources, such as the penetration of the signal into the surface snow, complex terrain and drastic changes in elevation and excessive interpolation (especially excessive extrapolation). But it is difficult to formally quantify and account for each of them.

And we have added these discussions in Section 4.4 in the revised manuscript.

(4) The description and names of many variables in the NetCDF file contain spelling errors, e.g., 'longitudt' should be longitude; 'latitudt' should be latitude; 'degress_north' should be 'degrees_north'; 'Baisn Num' should be 'Basin Num' and is based on Zwally et al., 2012 Antarctic and Greenland Drainage Systems. None of these affect the use of the data, but there are too many typos in the meta data.

Answer: Thank you for the comment. We have made relevant revision and updated it at http://dx.doi.org/10.11888/Glacio.tpdc.271658.

**Methodology description needs to be improved.**

Section 2.3 largely builds on the authors' previous work on the Antarctica Ice Sheet and published in Remote Sensing. Yet, I found this subsection difficult to read, largely due to the repeated use of same or very similar letter symbols in eqs. 1-5 but they actually have different meanings. Some terms are not introduced at all, such as (-1)AD in eq. 1 and (-1)im in eq. 2. This section needs to be rewritten to improve its clarity.

Answer: Thank you for the comment. We have rewritten it in the revised manuscript.

Section 2.4 and result parts related to EOF: this is relatively new in this work and needs further analyses and discussion in terms of the validity of assumptions, quality of the interpolated results. For instance, the assumption behind EOF is that the spatial patterns of elevation changes "are stationary in time" (Line 53), which do not hold as evident in Figures 4 and 5 and

numerous studies. How would temporal variations of spatial patterns, esp. those during the ERS period and the later Envisat-CryoSat-2 period, affect the EOF reconstruction?

Answer: Thank you for the comment. On the one hand, the assumption is the basis of using EOF to separate periodic changes of different frequencies of SECs. Similar to sea surface temperature and sea level change, SEC is the result of superposition of different periodic changes caused by different influencing factors (such as, temperature, wind, precipitation, ENSO, NAO, IPO, …). Many scholars have proved that the mass and elevation changes of Greenland ice sheet have significant periodic changes driven by climate (Bergmann et al., 2012; Slater et al., 2021; Zwally and Jun, 2002). Generally, spatial patterns of these periodic changes are considered to be stationary in time. EOF and other similar methods (e.g. singular value decomposition, Multi-channel Singular Spectral Analysis) have been used in the study of ice sheet change (Mémin et al., 2015; Zhang et al., 2019). In addition, EOF have been used for the reconstruction of historical sea surface temperatures (Smith et al., 1996), and sea level change (Chambers et al., 2002; Church et al., 2004; Jin et al., 2012). Therefore, in this study, we used it to reconstruct the surface elevation series over Greenland. On the other hand, in fact, only a small part of the grid cells need to be interpolated by EOF reconstruction. Most of the grid cells have elevation anomalies, and they are used in EOF reconstruction to minimize the interpolation (reconstruction) error using a linear least-squares estimator. In addition, if we regard the spread of thinning further inland not as a short-term acceleration thinning, but as part of a long-periodic change, it can actually be regarded as a steady-state spatial change. To sum up, the assumption is reasonable and tenable. The EOF reconstruction is suitable for interpolation of SE time series.

It is not clear to me what is the authors' basis for claiming "the superiority" of EOF over kriging (Line 73-74). My concern, as raised above, is unreliable EOF reconstruction and the large values along the coast.

Answer: Thank you for the comment. There are many outliers in the observations of early altimetry missions ERS-1 and ERS-2, so that there are no accurate observations available in many places, especially in areas with complex terrain and drastic changes in elevation along the ice sheet margin. As mentioned above, EOF reconstruction can be used for interpolation of SE time series. The purpose of solving EOF modes is to supplement the sparse monthly gridded data attributable to poor observations in the early years. In Fig. 3 of this study, we have compared the performance of Kriging interpolation and EOF reconstruction. In the Fig. A1 and A2, observation, Ordinary Kriging interpolation, and EOF reconstruction interpolation from August to December in 1991 and from January to June in 1994 are shown. It can be clearly seen that there are significant outliers caused by excessive extrapolation in Ordinary Kriging, while they are much less in EOF interpolation.

[Figure]

Fig. A1. Interpolation performance of EOF reconstruction and ordinary kriging: observation (upper),
Ordinary Kriging interpolation (middle), and EOF reconstruction interpolation (bottom) from August
to December, 1991.

[Figure]

Fig. A2. Interpolation performance of EOF reconstruction and ordinary kriging: observation (upper), Ordinary Kriging interpolation (middle), and EOF reconstruction interpolation (bottom) from January to June, 1994.

In the volumetric time series (Figures 6-8), it would be helpful to add the ones based on the CCI products with associated uncertainties as another cross-validation.

Answer: Thank you for the suggestion. If the volume change time series can be cross-validated with CCI products, it will be very helpful. Unfortunately, CCI only publicly released elevation change products, which are used in this study. Their elevation time series dataset that is used to derive volume change time series is not publicly available online. Hope we can do it in the future.

Figure 8c shows a sharp volumetric increase at the beginning of the time series (more than 300 km3 from 1992 to 1993). This doesn't seem to be correct. Could you double-check or validate with independent data?

Answer: Thank you for the comment. We have to admit that this is probably the outlier caused by the extrapolation error. From Fig. A1, we can see that the coverage of observations was very limited in 1991. Using very limited observations to generate grid, we must rely on more other information, such as the spatial variation information of signals with different frequencies derived by EOF. However, extrapolation is often accompanied by errors, including the EOF reconstruction. It does need more validation with other independent data. But we haven't collected these data yet. We have also put the non-interpolated SE anomalies in the dataset, hoping that any scholars could have better method to interpolate these monthly grid in the early years.

**Minor comments:**

It is important to mention EOF in the abstract.

Answer: Thank you for the suggestion. We have added it in the revised manuscript.

Line 48: specify what kind of data

Answer: Thank you for the suggestion. It is the surface elevation observations, and we have revised it in the revised manuscript.

Line 57: 'Therefore …'; I don't see a causal link between the limitations of the other approaches and the solution to be offered by data combination. Or is something missing in this last sentence of the paragraph?

Answer: Thanks for pointing out this issue. According to the suggestions of the Anonymous Referee #3, we have made removed it in the revised manuscript.

Line 93-95 fit better in methodology.

Answer: Thank you for the suggestion. We have moved it in Section 2.3 (Generation of surface elevation time series).

Line 208: change 'derived from' to 'caused by'

Answer: Thank you for the suggestion. We have revised it.

Line 227: 'Greenaldn' should be 'Greenland

Answer: Thanks for pointing out this issue. We have revised it.

Line 298: what is 'official relocation'?

Answer: Thanks for pointing out this issue. We have rewritten it as "surface elevation observations from data products have been relocated by the point of closest approach were used to suppress this influence" in the revised manuscript.

**References cited in authors' response:**

Bergmann, I., Ramillien, G., and Frappart, F.: Climate-driven interannual ice mass evolution in Greenland, Global Planet Change, 82-83, 1-11, https://doi.org/10.1016/j.gloplacha.2011.11.005, 2012.
Chambers, D. P., Mehlhaff, C. A., Urban, T. J., Fujii, D., and Nerem, R. S.: Low-frequency variations in global mean sea level: 1950–2000, Journal of Geophysical Research: Oceans, 107, 1-1-1-10, doi:10.1029/2001JC001089, 2002.
Church, J. A., White, N. J., Coleman, R., Lambeck, K., and Mitrovica, J. X.: Estimates of the Regional Distribution of Sea Level Rise over the 1950–2000 Period, J Climate, 17, 2609-2625, 10.1175/1520-0442(2004)017<2609:EOTRDO>2.0.CO;2, 2004.

Jin, T., Li, J., Jiang, W., and Chu, Y.: Low-frequency sea level variation and its correlation with climate events in the Pacific, Chinese Science Bulletin, 57, 3623-3630, doi:10.1007/s11434-012-5231-y, 2012.

Mémin, A., Flament, T., Alizier, B., Watson, C., and Rémy, F.: Interannual variation of the Antarctic Ice Sheet from a combined analysis of satellite gravimetry and altimetry data, Earth and Planetary Science Letters, 422, 150-156, https://doi.org/10.1016/j.epsl.2015.03.045, 2015.

Schröder, L., Horwath, M., Dietrich, R., Helm, V., van den Broeke, M. R., and Ligtenberg, S. R. M.: Four decades of Antarctic surface elevation changes from multi-mission satellite altimetry, The Cryosphere, 13, 427–449, doi:10.5194/tc-13-427-2019, 2019.

Slater, T., Shepherd, A., McMillan, M., Leeson, A., Gilbert, L., Muir, A., Munneke, P. K., Noël, B., Fettweis, X., van den Broeke, M., and Briggs, K.: Increased variability in Greenland Ice Sheet runoff from satellite observations, Nature Communications, 12, 6069, 10.1038/s41467-021-26229-4, 2021.

Smith, T. M., Reynolds, R. W., Livezey, R. E., and Stokes, D. C.: Reconstruction of Historical Sea Surface Temperatures Using Empirical Orthogonal Functions, J Climate, 9, 1403-1420, doi:10.1175/1520-0442(1996)009<1403:ROHSST>2.0.CO;2, 1996.

Sørensen, L. S., Simonsen, S. B., Forsberg, R., Khvorostovsky, K., Meister, R., and Engdahl, M. E.: 25 years of elevation changes of the Greenland Ice Sheet from ERS, Envisat, and CryoSat-2 radar altimetry, Earth and Planetary Science Letters, 495, 234–241, doi:10.1016/j.epsl.2018.05.015, 2018.

Zhang, B., Liu, L., Khan, S. A., van Dam, T., Bjørk, A. A., Peings, Y., Zhang, E., Bevis, M., Yao, Y., and Noël, B.: Geodetic and model data reveal different spatio-temporal patterns of transient mass changes over Greenland from 2007 to 2017, Earth and Planetary Science Letters, 515, 154-163, https://doi.org/10.1016/j.epsl.2019.03.028, 2019.

Zwally , J. H. and Jun, L.: Seasonal and interannual variations of firn densification and ice-sheet surface elevation at the Greenland summit, J Glaciol, 48, 199-207, 10.3189/172756502781831403, 2002.

**Reviewer 3:**

**Comments**

General comments:

This study describes the methods used to produce a new dataset of surface elevation changes of the Greenland Ice Sheet from satellite radar altimetry observations acquired by ERS-1, ERS-2, Envisat and CryoSat-2 spanning the period 1991 to 2020 and features some illustrations of this dataset over specific glaciers of the Greenland Ice Sheet. The main novelty in the methods described here is the use of EOF modes to fill in the monthly grids of surface elevation change.

I have some important general comments and questions that need to be addressed before I can recommend this manuscript for publication:

Is it a reasonable assumption to assume that 'spatial patterns of GrIS SECs are stationary in time' (L153) in your EOF reconstruction? The pattern of SECs has evolved from the 1990s to the 2003-2020 period. The authors themselves show this in Figure 4 where we can see the spread of thinning further inland between panels a (1991-2000) and c (2011-2020). How robust is this assumption?

Answer: Thank you for the comment. On the one hand, the assumption that "spatial patterns of GrIS SECs are stationary in time" is the basis of using EOF to separate periodic changes of different frequencies of SECs. Similar to sea surface temperature and sea level change, SEC is the result of superposition of different periodic changes caused by different influencing factors (such as, temperature, wind, precipitation, ENSO, NAO, IPO, …). Many scholars have proved that the mass and elevation changes of Greenland ice sheet have significant periodic changes driven by climate (Bergmann et al., 2012; Slater et al., 2021; Zwally and Jun, 2002). Generally, spatial patterns of these periodic changes are considered to be stationary in time. EOF and other similar methods (e.g. singular value decomposition, Multi-channel Singular Spectral Analysis) have been used in the study of ice sheet change (Mémin et al., 2015; Zhang et al., 2019). In addition, EOF have been used for the reconstruction of historical sea surface temperatures (Smith et al., 1996), and sea level change (Chambers et al., 2002; Church et al., 2004; Jin et al., 2012). Therefore, in this study, we used it to reconstruct the surface elevation series over Greenland. On the other hand, in fact, only a small part of the grid cells need to be interpolated by EOF reconstruction. Most of the grid cells have elevation anomalies, and they are used in EOF reconstruction to minimize the interpolation (reconstruction) error using a linear least-squares estimator. In addition, if we regard the spread of thinning further inland not as a short-term acceleration thinning, but as part of a long-periodic change, it can actually be regarded as a steady-state spatial change. To sum up, the assumption is reasonable and tenable.

 The SECs dataset comes with an uncertainty estimation, which is great but I am not convinced by the uncertainty assessment made here. I understand that using the MAD is straightforward but I wonder if it is a good measure of the uncertainty of the measurement. While I recognise that it is difficult to formally account for the different sources of uncertainty in the altimetry SECs measurements, some useful information/discussion could be added regarding the uncertainty assessment – for instance a discussion of which step in your processing contributes to larger uncertainties (is it the inter-mission calibration, the formulation-n of the plane-fit model…?)

Answer: Thank you for the comment. Generally, previous studies (Schröder et al., 2019; Paolo et al., 2016; Sørensen et al., 2018; Shepherd et al., 2019) often used standard deviation as the uncertainty of elevation time series. In this study, we used a median filter to yield a more robust solution for each grid cell. To get a consistent estimator similar to the standard deviation, we use the scaled MAD to estimate the uncertainty of elevation time series. It is indeed very difficult to formally account for the different sources of uncertainty in the altimetry SECs measurements. Meanwhile, it is difficult to evaluate the uncertainty caused by each step of processing, too. Thus, we have only added some qualitative discussions on the evaluation uncertainty in Section 4.4 Limitations of the merged surface elevation time series.

I would rename Section 4 'Comparison to Independent Datasets' as this section is not really an uncertainty assessment but more a validation/comparison analysis to the ATM and CCI datasets. I suggest to move the first subsection 'Error sources' at the end of this section and rename it 'Limitations of the dataset' or something along those lines.

Answer: Thank you for the suggestion. We have made relevant revision in the revised manuscript.

In the 'Results' section on the analysis surface elevation anomaly time-series, the authors look at decadal trends in SECs over specific glaciers which highlight the long record that they have produced. This is a good illustration of the potential use of the dataset and it would be good to also feature an illustration of the ability of the dataset to look at seasonal changes in elevation change to highlight the high temporal resolution of their dataset, or at least to discuss this in the text.

Answer: Thank you for the suggestion. We have made relevant revision in the revised manuscript.

Do you see any step change in the SEC time-series following the 2012 extreme melt event in the interior of the ice sheet? Could you comment on whether this artefact is present in your time-series or if your processing scheme is able to correct for this effectively?

Answer: Thank you for the comment. The step change in the SEC time-series in the interior the GrIS during the extreme melt event in July 2012 was resulting from the change of penetration depth caused by surface melting. We used elevations retracked by threshold offset center of gravity retracker (ICE-1 re-tracker and OCOG re-tracker) and the common strategy of including corrections for waveform parameters into the least-squares regression model (see Eq. (1)) to mitigate the time-variable penetration effects of the radar signal. Because the threshold offset center of gravity retracker is less sensitive to changes in volume scattering, it has been used to reduce the effect of penetration (Nilsson et al., 2015; Schröder et al., 2017; Schröder et al., 2019). The latter has also been performed in many previous studies (Flament and Remy, 2012; Sørensen et al., 2018; Zhang et al., 2020). However, as presented by Slater et al. (2019), the influence of the time-variable penetration depth would not be completely eliminated, even applying a waveform deconvolution procedure(Mcmillan et al., 2016). Thereby, a weak signal of artefact step increase caused by the melting event in 2012 might be found in the merged time series. In the future, with the accumulation of long-term continuous observations by satellite laser altimetry ICESat-2, it seems feasible to obtain actual penetration depth and model predictions to better compensate for the fluctuations in penetration depth. On the bright side, surface penetration suppresses noise induced by seasonal snowfall, making radar altimetric measurements more relevant to mass change than those obtained from laser altimetry(Sørensen et al., 2018). Therefore, our multiple radar altimetry missions SE time

series is more suited to track dynamical processes and inter-annual or long-term surface processes(Zhang et al., 2020; Simonsen et al., 2021). We have added the discussion in the Section 4.4 in the revised manuscript.

It is straightforward to download the data from the link provided but there are quite a lot of typos in the metadata of the NetCDF file: 'latitudt', 'longitudt' or for instance the description of the basins variable states: 'Antarctic_Drainage_System_Boundaries_and_Masks' when it should be 'Greenland_Drainage_System_Boundaries_and_Masks'.

Answer: Thank you for the comment. We have made relevant revision and updated it at http://dx.doi.org/10.11888/Glacio.tpdc.271658.

I made some specific comments and suggestions below, which I hope will help improve this paper.

**Specific comments:**

L8: 'for study of ice sheet variation and its response to climate change', please reformulate

Answer: Thanks for pointing out this issue, and we have rewritten it in the revised manuscript.

L26: You also need a density model for the snow and firn layer in addition to a model of the distribution of the ice layers within the firn column to convert volume change to mass change. Please clarify this sentence.

Answer: Thanks for pointing out this issue, and we have rewritten it in the revised manuscript.

L28: Please be more specific in this sentence, a long-term time series of EC is essential to assess the impact of climate change on the ice sheets rather than 'to assess climate change directly'.

Answer: Thanks for pointing out this issue, and we have made relevant revision.

L56: I recall that the deconvolution method from Slater et al (2019) does provide time-variable penetration depth over the interior of the Greenland Ice Sheet.

Answer: Thank you for the comment. Slater et al. (2019) did estimate the penetration depth of CryoSat-2 LRM observations over the region of the Greenland ice sheet interior above the 2000-m using deconvolution. However, the deconvolution is based on the assumption that the effects of large-scale surface slope and footprint-scale topographic undulations upon the waveform shape are negligible and is therefore only appropriate in areas of flat terrain. In fact, for SARIn observations, Slater et al. (2019) used the waveform parameters (backscattered power) to account for temporal variations in the degree of radar penetration, too. It implies that Slater et al. (2019) also agree using waveform parameters to mitigate time-variable penetration effects. In addition, they only said that "the penetration variation can be compensated effectively by incorporating the deconvolution penetration depth into the surface height retrieval", not that "the penetration variation can be compensated completely by incorporating the deconvolution penetration depth into the surface height retrieval".

L57-58: 'a more reasonable approach' than? I would argue that the reason to combine data from several radar altimetry missions is that it is the only way to get a long record of surface

elevation changes. I would remove this sentence, as it doesn't link with what was said in the paragraph.

Answer: Thanks for pointing out this issue, and we have made removed it in the revised manuscript.

L59: By irregular, do you mean that the satellite tracks deviate from the ground-tracks? I think you need to be a bit more specific here as this sentence could be misread. ERS-1/2 and ENVISAT have a repeat cycle and CryoSat-2 a drifting orbit but all missions have regular a ground track pattern.

Answer: Thanks for pointing out this issue. We mean that the coverage of ground tracks of polar orbiting altimetry satellites over the polar ice sheets is uneven. And we have made relevant revision.

L64: 'for to unobserved grid cells'

Answer: Thanks for pointing out this issue, and we have made relevant revision.

L85: Specify how similar the sensors are (for instance they all use Ku-band etc). You could also mention Sentinel-3 here. Could your processing scheme be applied to Sentinel-3 data as well?

Answer: In this study the similar sensors is mean similar Ku-band altimeters. We have specified it in the revised manuscript. The similar sensors used here is mainly to avoid the influence of the difference in penetration depth of signals of different frequencies. In theory, our processing scheme can also be applied to Sentinel-3 data. If the penetration of the radar signals can be handled well, our processing scheme can also be used for combining observations from laser altimeters (ICESat and ICESat-2) and radar altimeters (including SARAL, Sentinel-3 and others) well.

L93: typo 'SARIn'

Answer: Thanks for pointing out this issue, and we have made relevant revision.

L171-172: How do you estimate the seasonal signals here? Do you use the terms from the plane fit model or do you estimate the seasonal components using a time-series decomposition technique?

Answer: Thank you for the comment. We use a least-squares fitting model with a second-order polynomial and seasonal terms to estimate the seasonal signals. And we have added it in the revised manuscript.

L175: Can you add the average proportion of the monthly grid that has be filled in using the EOF modes for each mission? Data are rather sparse during the ERS-1/2 periods compared to the CryoSat-2 period when only small gaps between tracks occur. It would be useful to add a sentence in the text to reflect this.

Answer: Thank you for the suggestion. We have added it in the revised manuscript.

L204: Did you look at trends in surface mass balance to support this claim?

Answer: Thank you for the comment. This claim is not made by this study, but the result of other studies (Mouginot et al., 2015; Aschwanden et al., 2016; Shepherd et al., 2020; Wood et al., 2021). Both trends in surface mass balance and in ice discharge have been carefully examined by them. The accelerated and expanded thinning in many outlet glaciers found from our merged time series is in line with their claim. It implies that our time series is reliable.

L217: Are there limitations of your dataset to look at SECs over some small glaciers? Is there an optimal area size for glaciers for which your dataset would be the most useful? Is your dataset reliable close to the termini of glaciers where altimetry measurements are usually sparse and the slope is high?

Answer: Thanks for pointing out this issue. We agree with your comment. Due to the excessive size of footprint of radar altimeter, surface elevations from radar altimeter in the steeper and active areas would have great uncertainties. It is a natural defect of radar altimeter. To overcome this problem, relocation by the point of closest approach and iterative calculation of least-squares regression have been done in this study. Meanwhile, we estimate the uncertainty of each gridded surface elevation anomaly. Although we believe that each estimated elevation anomaly is a reasonable solution obtained from the altimetry observations, we have to admit that if the terrain is too steep or the glacier is too small, our results might have great uncertainty. However, it is difficult to determine a threshold of slope and area to determine which glaciers our time series is applicable to. It requires more external data to determine them. We will continue the job in the future. Sørensen et al. (2018) has arbitrarily excluded all grid cells which are located on slopes exceeding 1.5˚ to avoid the possible large uncertainty. We have made relevant explanations in the revised manuscript.

L227: typo 'Greenland Ice Stream'

Answer: Thanks for pointing out this issue, and we have made relevant revision.

L235-236: Not all fluctuations in SECs are caused by climate change, in case of short-term fluctuations it is hard to distinguish between natural climate variability and climate change

Answer: Thank you for the comment. We have removed "short-term fluctuations" in the revised manuscript.

L292: Here the advantage is to use radar altimetry instead of laser altimetry to look at elevation changes induced by SMB processes. I would add in the sentence 'by combining data from multiple radar altimetry missions'

Answer: Thank you for the suggestion. We have added it in the revised manuscript.

L553: typo 'left-hand maps'

Answer: Thanks for pointing out this issue, and we have made relevant revision in the revised manuscript.

Fig 9: Please clarify the caption, it's unclear from the caption what is shown on Figure 9b. Reading the caption alone, it looks like maps a and b are showing the same thing. Also, what do you by the same epochs for map c?

Answer: Thank you for the suggestion. We have modified the caption of Fig .9.

Table 2: What do you mean by 'periods across the Envisat and CryoSat-2 connections?' I would maybe say 'overlap' rather than 'connections'.

Answer: Thank you for the suggestion. We agree that 'overlap' is better, and we have made relevant revision in the revised text.

**References cited in authors' response:**

Aschwanden, A., Fahnestock, M. A., and Truffer, M.: Complex Greenland outlet glacier flow captured, Nature Communications, 7, 10524, doi:10.1038/ncomms10524, 2016.

Bergmann, I., Ramillien, G., and Frappart, F.: Climate-driven interannual ice mass evolution in Greenland, Global Planet Change, 82-83, 1-11, https://doi.org/10.1016/j.gloplacha.2011.11.005, 2012.

Chambers, D. P., Mehlhaff, C. A., Urban, T. J., Fujii, D., and Nerem, R. S.: Low-frequency variations in global mean sea level: 1950–2000, Journal of Geophysical Research: Oceans, 107, 1-1-1-10, doi:10.1029/2001JC001089, 2002.

Church, J. A., White, N. J., Coleman, R., Lambeck, K., and Mitrovica, J. X.: Estimates of the Regional Distribution of Sea Level Rise over the 1950–2000 Period, J Climate, 17, 2609-2625, 10.1175/1520-0442(2004)017<2609:EOTRDO>2.0.CO;2, 2004.

[revised manuscript text omitted]

Zhang, B., Liu, L., Khan, S. A., van Dam, T., Bjørk, A. A., Peings, Y., Zhang, E., Bevis, M., Yao, Y., and Noël, B.: Geodetic and model data reveal different spatio-temporal patterns of transient mass changes over Greenland from 2007 to 2017, Earth and Planetary Science Letters, 515, 154-163, https://doi.org/10.1016/j.epsl.2019.03.028, 2019.

Zwally , J. H. and Jun, L.: Seasonal and interannual variations of firn densification and ice-sheet surface elevation at the Greenland summit, J Glaciol, 48, 199-207, 10.3189/172756502781831403, 2002.

---

## Referee Report (RR1)

I'm satisfied with the response provided by the authors and I think that the manuscript has clearly been improved. There is now a good discussion of the strengths and limitations of this new dataset, which I think will be greatly appreciated by future users. The metadata have also been corrected. I made a few minor comments below (the line numbers refer to the line numbers of the track-changed manuscript):

**L12:** Please state concisely what 'The sophisticated corrections' consist of here. This is too vague for an abstract.

**L124-127:** You need to justify your choice of retracker here. You spend some time in the introduction (L56-61) to mention three different techniques to mitigate the effects of radar penetration so I think it would be nice to reflect on that and state what method you chose and why.

**L162:** Do you have enough data within a 2 km grid cell to constrain the least-square fit during the ERS-1/2 missions?

**L163:** What ice sheet mask/delineations are you using? Please specify here whether you're using Rignot's, Zwally's  definition or something else.

**L178:** I would add 'at least 100 elevation anomalies in the 216 months of the 2003-2020 period are retained' for clarity

**L191:** 'and then add them back to the EOF reconstruction results' instead of 'return them'

**L199:** 'can be calculated'

**L267:** 'Ice velocity'

**L277-279:** I suggest moving this sentence at the end of section 2.4 as it belongs more to the methodology than the results section.

**L398:** 'even when applying'

**L399:** I would be more specific 'a small residual signal caused by the 2012 melt event and manifesting as a surface elevation increase signal is found in the merged time-series'. Can you quantify this elevation step in your time-series to give the user an indication of how small the signal is? You could calculate the elevation difference before/after summer 2012 for the ice sheet as a metric.

---

## Author Response (AR2)

Dear the reviewers:

First of all, we would like to take this opportunity to thank the reviewer for your constructive comments and relevant questions. By adding the answers/revisions to these questions to the revised version of the manuscript, we feel that the quality of the manuscript has been improved. A revised manuscript has been submitted, and all of corrections/modifications are only included in the revised manuscript for the sake of non-repeat. Extra answers to your concerns and questions are presented as follows.

**Reviewer 1:**

**Comments**

Review of Zhang et al.

I'm satisfied with the response provided by the authors and I think that the manuscript has clearly been improved. There is now a good discussion of the strengths and limitations of this new dataset, which I think will be greatly appreciated by future users. The metadata have also been corrected. I made a few minor comments below (the line numbers refer to the line numbers of the track-changed manuscript):

L12: Please state concisely what 'The sophisticated corrections' consist of here. This is too vague for an abstract.

Answer: Thanks for pointing out this issue. The corrections are intermission bias corrections. We shouldn't use sophisticated here. We have made relevant revision in the revised manuscript.

L124-127: You need to justify your choice of retracker here. You spend some time in the introduction (L56-61) to mention three different techniques to mitigate the effects of radar penetration so I think it would be nice to reflect on that and state what method you chose and why.

Answer: Thank you for the suggestion. We have justified in the revised manuscript.

L162: Do you have enough data within a 2 km grid cell to constrain the least-square fit during the ERS-1/2 missions?

Answer: Sorry to mislead you. In this study, the least-squares fitting was performed on a 2 km polar-stereographic grid, but not within a 2 km grid cell. For each grid node, all observations within 2.5 km of the centre of the grid node are used for the iterative least-squares estimation. This can ensure that the most grid node have enough data to constrain the least-squares fitting during each mission, including the ERS-1 and ERS-2 missions. We have made relevant revision in the revised manuscript.

L163: What ice sheet mask/delineations are you using? Please specify here whether you're using Rignot's, Zwally's definition or something else.

Answer: Thank you for the suggestion. In this study, we used the Zwally's ice sheet mask. We have specified it in the revised manuscript.

L178: I would add 'at least 100 elevation anomalies in the 216 months of the 2003-2020 period are retained' for clarity

Answer: Thank you for the suggestion. We have added it in the revised manuscript.

L191: 'and then add them back to the EOF reconstruction results' instead of 'return them'

Answer: Thank you for the suggestion. We have made relevant revision in the revised manuscript.

L199: 'can be calculated'

Answer: Thanks for pointing out this issue, and we have made relevant revision in the revised manuscript.

L267: 'Ice velocity'

Answer: Thanks for pointing out this issue, and we have made relevant revision in the revised manuscript.

L277-279: I suggest moving this sentence at the end of section 2.4 as it belongs more to the methodology than the results section.

Answer: Thank you for the suggestion. We have moved it at the end of section 2.4 in the revised manuscript.

L398: 'even when applying'

Answer: Thanks for pointing out this issue, and we have made relevant revision in the revised manuscript.

L399: I would be more specific 'a small residual signal caused by the 2012 melt event and manifesting as a surface elevation increase signal is found in the merged time-series'. Can you quantify this elevation step in your time-series to give the user an indication of how small the signal is? You could calculate the elevation difference before/after summer 2012 for the ice sheet as a metric.

Answer: Thank you for the suggestion. We have estimated that the mean elevation difference before/after summer 2012 for the regions above 2000 m in altitude is about 0.16 m between the months before (January–June, 2012) and after (August–December, 2012) the extreme melt event, which is consistent with Slater et al. (2019) of $0.21 \pm 0.09$ m. And we have made relevant revision in the revised manuscript.

**References cited in authors' response:**

Slater, T., Shepherd, A., Mcmillan, M., Armitage, T. W. K., Otosaka, I., and Arthern, R. J.: Compensating Changes in the Penetration Depth of Pulse-Limited Radar Altimetry Over the Greenland Ice Sheet, IEEE Transactions on Geoscience and Remote Sensing, 57, 9633-9642, doi:10.1109/TGRS.2019.2928232, 2019.

---

## Author Response (AR3)

Dear the reviewers and editor:

First of all, we would like to take this opportunity to thank the reviewer for your constructive comments and relevant questions. By adding the answers/revisions to these questions to the revised version of the manuscript, we feel that the quality of the manuscript has been improved. A revised manuscript has been submitted, and all of corrections/modifications are only included in the revised manuscript for the sake of non-repeat. Extra answers to your concerns and questions are presented as follows.

**Editor**

**Comments**

Comments to the author:

I am glad to see your revised and improved manuscript. Please provide some descriptions of dataset in section 5, such as data format, organization, and so forth, which will help users for open and understand the dataset.

Answer: Thanks for pointing out this issue. And we have made relevant revision in the revised manuscript.